# Self-shrinking soft demoulding for complex high-aspect-ratio microchannels

Dongliang Fan [1,2], Xi Yuan [3,4], Wenyu Wu [5], Renjie Zhu [1,2], Xin Yang [1,2], Yuxuan Liao [1,2], Yunteng Ma [1], Chufan Xiao [3,4], Cheng Chen [6], Changyue Liu [3,4], Hongqiang Wang [1,2,7] ✉ & Peiwu Qin [3,4] ✉

Microchannels are the essential elements in animals, plants, and various artificial devices such as soft robotics, wearable sensors, and organs-on-a-chip. However, three-dimensional (3D) microchannels with complex geometry and a high aspect ratio remain challenging to generate by conventional methods such as soft lithography, template dissolution, and matrix swollen processes, although they are widespread in nature. Here, we propose a simple and solvent-free fabrication method capable of producing monolithic microchannels with complex 3D structures, long length, and small diameter. A soft template and a peeling-dominant template removal process are introduced to the demoulding process, which is referred to as soft demoulding here. In combination with thermal drawing technology, microchannels with a small diameter (10 μm), a high aspect ratio (6000, length-to-diameter), and intricate 3D geometries are generated. We demonstrate the vast applicability and significant impact of this technology in multiple scenarios, including soft robotics, wearable sensors, soft antennas, and artificial vessels.

Natural microscale vessels ubiquitously exist in animals and plants since they are critical for nutrient transportation and byproduct removal[1–3]. In recent decades, the artificial counterparts, namely microchannels, have been among the most quickly emerging and widely spreading technologies in diverse ranges of disciplines and contexts, including drug discovery[4], biomedical studies[4,5], chemical analysis[6], and most recently, soft robotics[7–9], wearable sensors[10,11], and artificial vessels[5,12,13]. For example, high-aspect-ratio channels endowed soft actuators with great entanglement for grasping[9,14], and complex 3D optical laces were able to mimic the afferent sensory neural network[15]. High-aspect-ratio microchannels with 3D geometries are critical to improving particle sorting efficiency[16] and reappeared

alveoli's function[17]. However, compared to the natural micro-vessels, the creation of artificial microchannels is still challenging due to their topological complexity and size. Researchers have only achieved either ultrathin channels or complex 3D structures[16,18], while nature generates intertwined vessels varying vastly in diameter, shape, and 3D structure.

The widely accepted soft lithography technique suffers from limited cross-sectional shapes (rectangular) and spatial structures (two-dimensional (2D) patterns only), intensive labour, and expensive fabrication devices, and it is unable to generate monolithic structures[6,19]. Emerging methods, such as additive manufacturing[17,20,21], matrix swollen[16,22–24], and template dissolution[12,13,16,18,25–27] can hardly

[1]Shenzhen Key Laboratory of Biomimetic Robotics and Intelligent Systems, Department of Mechanical and Energy Engineering, Southern University of Science and Technology, Shenzhen, Guangdong 518055, China. [2]Guangdong Provincial Key Laboratory of Human-Augmentation and Rehabilitation Robotics in Universities, Southern University of Science and Technology, Shenzhen 518055, China. [3]Institute of Biopharmaceutical and Health Engineering, Tsinghua Shenzhen International Graduate School, Shenzhen, Guangdong 518055, China. [4]Center of Precision Medicine and Healthcare, Tsinghua-Berkeley Shenzhen Institute, Shenzhen, Guangdong Province 518055, China. [5]School of System Design and Intelligent Manufacturing, Southern University of Science and Technology, Shenzhen, Guangdong 518055, China. [6]Department of Biomedical Engineering, National University of Singapore, Singapore 117575, Singapore. [7]Southern Marine Science and Engineering Guangdong Laboratory (Guangzhou), Guangzhou 510000, China. ✉e-mail: wanghq6@sustech.edu.cn; pwqin@sz.tsinghua.edu.cn

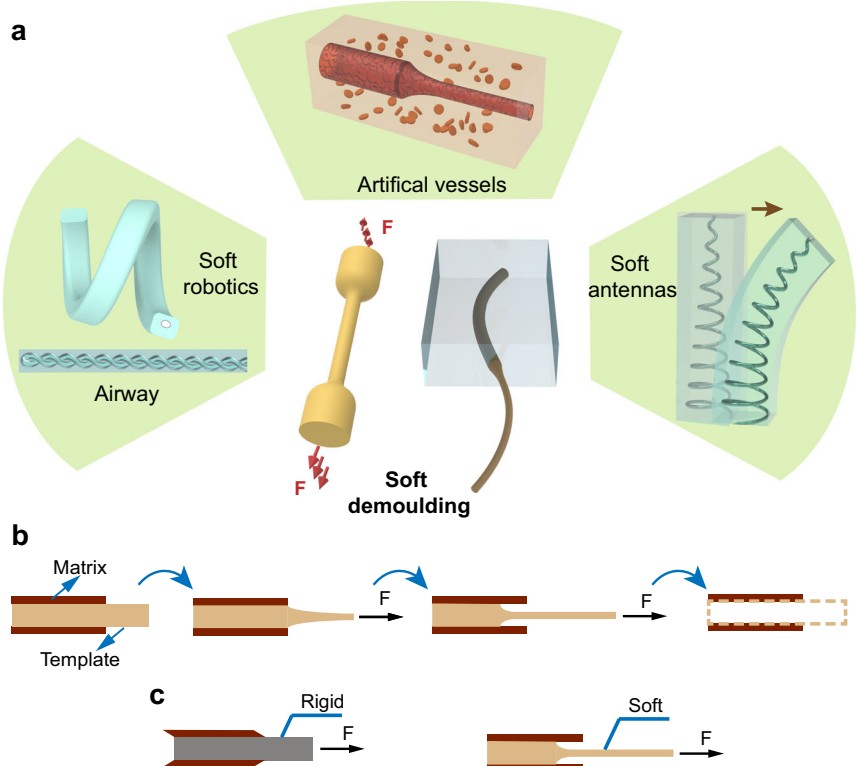

**Fig. 1 | The concept and mechanism of soft demoulding. a** The conceptual schematic of soft demoulding and its typical applications. **b** The soft demoulding process: the template is first embedded in the matrix, then it shrinks its cross-sectional area and is peeled out when an external force is applied, and consequently, the channel is formed. **c** The comparison of extraction of a rigid template (conventional method) and a soft template (proposed in this work), which are shear-dominated demoulding and peeling-dominated demoulding, respectively.

generate microchannels that are ultrathin, long (high aspect ratio), and complex in geometry with high efficiency. Additive manufacturing can generate 3D microchannels in intricate topological geometries, but the feature size and surface roughness are limited by the fabrication processes[17,20]. Matrix swollen methods require swelling and deswelling processes of matrices for templates demoulding, which cause buckling of the matrices and solvent residual[28]. Complex and ultra-thin microchannels can be fabricated by template dissolution methods, but dissolution and draining out become challenging due to the capillary effect when the channels are just tens of micrometres[18,29]. Other methods, such as employing liquid template[30] and laser processing technology[31], suffer limitations for 3D geometries and smooth channel generation. In addition, 3D microchannels assembly is challenging due to templates fixing and removal processes. Most of the current fabrication methods are inadequate for biological applications that strictly require nontoxic and biocompatible elements[18,22]. Hence, novel techniques that generate complex 3D structured, nontoxic, and slender monolithic microchannels are expected to revolutionize the vast applications where microchannels are indispensable.

Inspired by the tension-induced necking phenomenon during the cold drawing process of the polymeric specimens[32,33], we propose a simple, fast, solvent-free method to generate micro-level 3D monolithic channels (Supplementary Fig. 1). We employ a template that is softer than the matrix and stretch the soft template while it is embedded in the matrix, to remove the template from the matrix by a peel-dominant process (Fig. 1a, b). Here we call this methodology soft demoulding since the template shrinks itself during the extracting. This process requires a significantly lower force than the shear-dominant removal process that happens in the conventional rigid template extraction[7] (Fig. 1c and Supplementary Movie 1). By varying the dimensions and geometries of the template, various microchannels from one-dimensional (1D) to 3D can be created for different application scenarios (Fig. 1a).

## Results
### Fabrication of soft templates
The soft demoulding technique contains two main traits: a soft template and a gentle demoulding process. The soft template can be fabricated by various methods, including 3D printing[34], inkjet printing[21], and injection moulding[35], only if it is softer than the matrix. This work contrives the soft templates by thermal drawing[36]; i.e., dipping the tip of a thin rod into a polymer melt and then drawing the rod out of the melt (Fig. 2a). Attached to the rod tip, the polymer melt in the air transforms into the filament shape due to the viscosity and surface tension and solidifies quickly due to the temperature gradient. Various thermoplastic polymers are available and adaptable for this manipulation[37], including low-cost, vastly applied polyethylene and polyurethane. This fabrication method is fast, simple, and productive. The filament can be produced with a circular cross section and controllable diameter in an extensive range from tens of microns to hundreds of microns. The diameter of the filament $D$ is determined by[36]:

$$D = \frac{C}{\sqrt{v}} \tag{1}$$

where $C$ is a constant, and $v$ is the drawing speed (see "Methods" section 'Fabrication of soft templates' and Fig. 2b). The filament diameter is constant at the same drawing speed (Fig. 2c). Variable diameters, such as a tapered shape (Fig. 2d), can be generated on a single filament by simply altering the drawing speed.

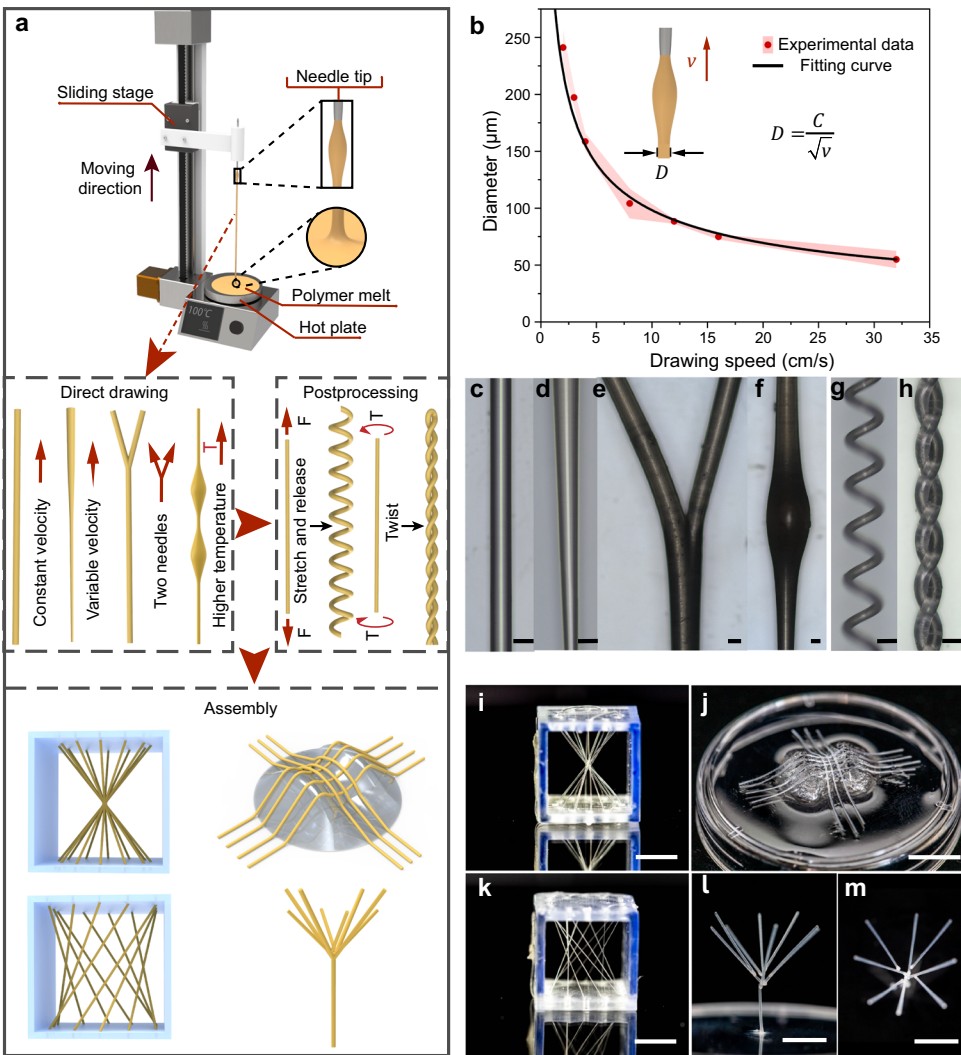

**Fig. 2 | Fabrication methods of soft templates and the resultant prototypes.**
**a** The fabrication methods for the soft templates, including direction drawing, postprocessing, and assembly. A series of shapes and geometries can be generated by varying the parameters in each step. **b** The relationship between filament diameter $D$ and drawing speed $v$ ($C$ is a constant). The red dots and pink cloud represent the mean values and standard deviations. **c–h** Soft templates of different shapes (i.e., a straight, a taper-shaped, a branched, a spindle-knotted, a helical, and a plectoneme structure) fabricated by direct drawing and postprocessing. Images in **c–h** are representative of five independent soft templates (experimental replicates). Scale bars, 100 μm. **i–m** Soft templates of 3D complex geometries (i.e., a conical surface, a saddle surface, a hyperboloid surface, and a tree-like structure) fabricated by assembly. Scale bars, 5 mm.

In addition to the 1D template, more complex profiles can be fabricated by tuning other fabrication parameters. For example, we generated a branched structure by pulling two needle tips in two directions (see Fig. 2e). At a higher heating temperature (130 °C in this work), we created a filament with a series of spindle-knotted structures on the filament (Fig. 2f), which results from the synergistic interaction of the viscosity and surface tension[38]. Via postprocessing, more complex spatial shapes were produced. For instance, a helical template was fabricated by further stretching the straight filament before it was thoroughly cooled down (Fig. 2g). Moreover, we generated a plectoneme structure by twisting both ends of the filament (Fig. 2h). By template arrangement and assembly, more intricate 3D structures were created, such as a conical surface, a saddle surface, a hyperboloid surface, and a tree-like structure (Fig. 2i–m). The surface of the template fabricated by this method is smoother (e.g., $S_a$ = 0.010 μm at Supplementary Fig. 2a) than those fabricated by other methods such as 3D printing ($S_a$ is more than 0.4 μm[39]) due to the fluidity of the produced material. A smooth surface can reduce the demoulding resistance force and leave a low roughness surface on the microchannels.

## Soft demoulding

Pulling out the template from the matrix is another grand challenge in forming a microchannel. With previously rigid templates[7], both the shear force and the template's fracture force determine the diameter of the channel (Fig. 3a–c and Supplementary Fig. 3). There is competition between the critical fracture force of the rigid template and the shear force during pulling. Once the shear force is larger than the critical fracture force, the template fractures and the demoulding fails. Only when the critical fracture force is larger, the template can be pulled out. Assuming the template is a simple straight wire with a round cross-section, the shear force $F_{shear}$ is determined by:

$$F_{shear} = \tau \pi d l \qquad (2)$$

where $\tau$ is the shear stress, $d$ is the filament diameter, and $l$ is the embedded length. The shear force increases linearly with the diameter and the embedded length of the template, as shown in Fig. 3b. The

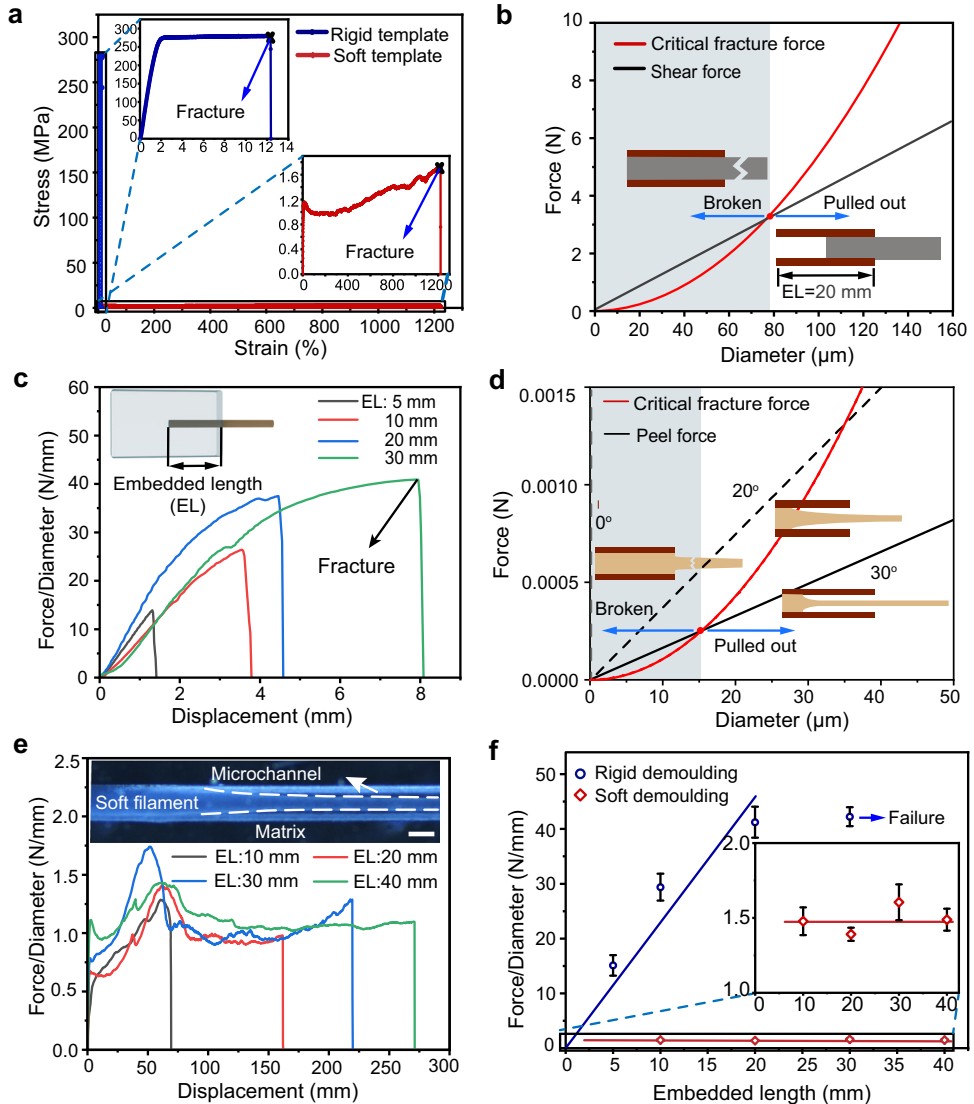

**Fig. 3 | Mechanisms of soft demoulding and rigid demoulding. a** The stress-strain curvatures of a rigid template (copper) and a soft template (thermoplastic resin). **b** The critical fracture force and shear force of the rigid template (copper wire with 20 mm embedded length) for different filament diameters. The shadow region represents the range of the demoulding failure. **c** The force/diameter–displacement curve with different embedded lengths (EL) for rigid templates (copper). The diameter of the copper wire is 80 μm. **d** The critical fracture force and peel force varying with the diameter of soft template

(thermoplastic resin). The intersections in the plot indicate the minimum diameter for initiating the peel process with certain peel angles (0, 20, and 30°). The shadow region represents the demoulding failure. **e** The pull-out force of the soft templates. Peeling of the soft template from the matrix is shown in the inset. Images in the inset are representative of three independent soft demoulding processes (experimental replicates). Scale bar, 100 μm. **f** Comparison of the pull-out force of the rigid templates and the soft templates. The data are presented as mean values ± standard deviation for the number of trials $n = 3$.

template's fracture force $F_{frac}$ is calculated as:

$$F_{frac} = \sigma_f A = \frac{\sigma_f \pi d^2}{4} \quad (3)$$

where $\sigma_f$ is the fracture stress, and $A$ is the cross-sectional area of the template. Hence, a longer embedded length demands a thicker diameter for successful extraction. To avoid fracture, we have $F_{shear} \le F_{frac}$, and thus:

$$\frac{l}{d} \le \frac{\sigma}{4\tau} \quad (4)$$

which indicates that the aspect ratio of the channel is intrinsically limited by the nature of the rigid template materials, i.e., the strength and adhesive energy density. For instance, the maximum aspect ratio

for the PDMS matrix and the copper template system is 258, according to Eq. (4) ($\sigma$ = 6300 MPa, $\tau$ = 6.1 MPa, based on experimental results in this work). As shown in Fig. 3c and Supplementary Fig. 3b, c, when the embedded length was 30 mm, the copper (diameter: 80 μm) and nylon template (diameter: 100 μm) ruptured due to the large aspect ratios (375 and 300, respectively). Such a strong force might break a fragile matrix (e.g., agarose gels). Moreover, a rigid template inevitably causes wear on the channel surfaces due to the rigidity of the template and the large shear force.

In contrast, for the soft demoulding method, peeling instead of shearing is dominant during the extraction process due to the larger stretching ratio of the template (the ultimate strain of the thermoplastic resin (1200%) was 100 times that of the copper wire (12%), as shown in Fig. 3a). The aspect ratio of the channel doesn't limit by the length because the large deformation of the soft template transfers the demoulding mechanism to a peeling process (see

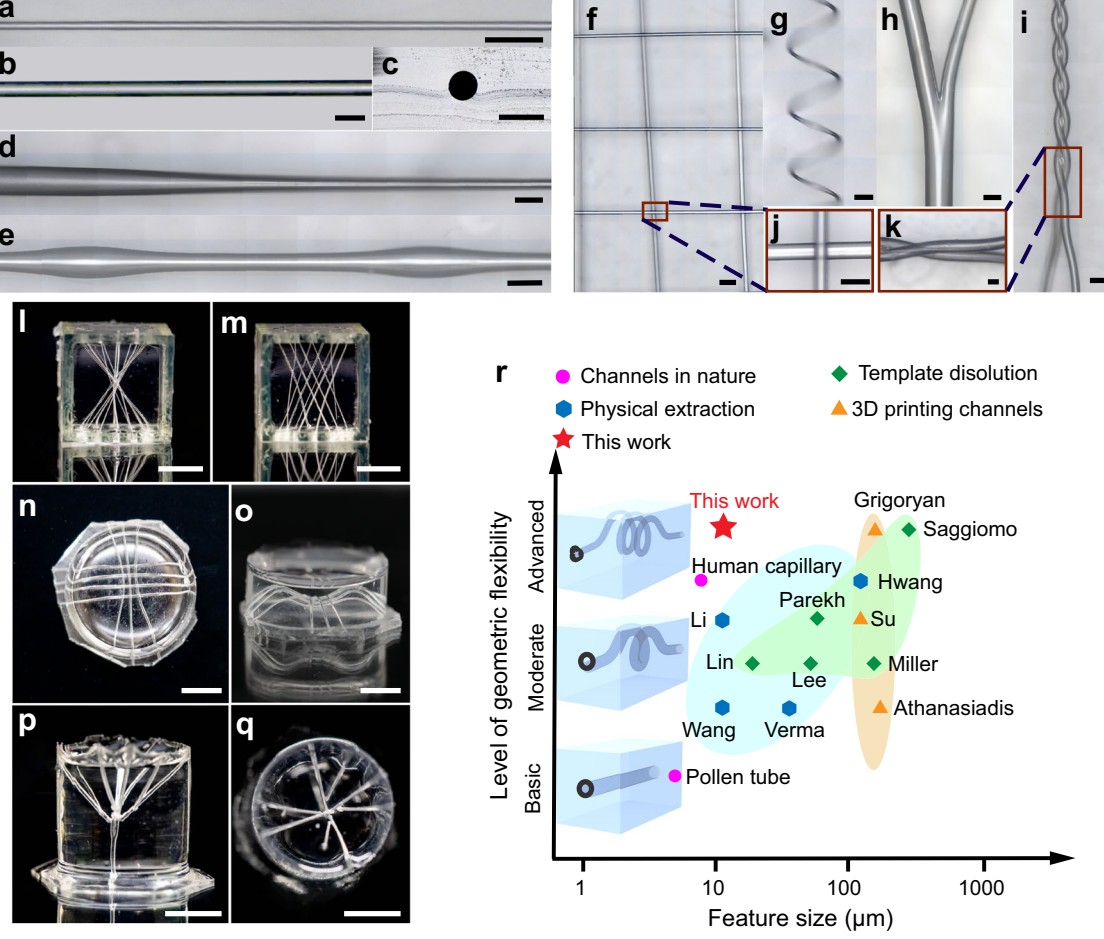

**Fig. 4 | Microchannel structures fabricated by soft demoulding. a** The thinnest channel (10 μm in diameter) fabricated in this work. Scale bar, 50 μm. **b**, **c** The microchannel and its circular cross-section. Scale bars, 50 μm. **d** The microchannel with tapered geometry. The diameter of the left side of the tapered channel is 250 μm, and the right-side diameter is 40 μm. Scale bar, 200 μm. **e** The microchannel with a spindle-knotted shape. Scale bar, 500 μm. **f** The microchannel lattice. Scale bar, 500 μm. **g** The helical microchannel. Scale bar, 200 μm. **h** The branched microchannel. Scale bar, 100 μm. **i** The microchannel with plectoneme structure. Scale bar, 500 μm. **j**, **k** are the feature parts of **f** and **i**, respectively. Scale bars, 200 μm. **l**–**q** The novel 3D microchannel structures. Both **n** and **o** show different views of the same prototype and the same to **p** and **q**. Scale bars, 5 mm. Images in **a**–**q** are representative of five independent microchannels (experimental replicates). **r** Feature size and geometric flexibility for microchannel fabrication studies (Assembly capability is not considered for comparison).

"Methods" section 'Fabrication of the microchannels by soft demoulding'), and the peel force has no relationship with the embedded length of the soft template (Fig. 3d, f). The peel force $F_{peel}$ for soft demoulding of thermoplastic resin (hot melt adhesive, 3748Q) can be expressed by ref. 40:

$$F_{peel} = \Delta E_S \frac{\pi d}{(1 - \cos\theta)} \qquad (5)$$

where $d$ is the filament diameter, $\theta$ is the peel angle, and $\Delta E_S$ is the adhesive energy. The peel force remained consistent for the samples with different embedded lengths (10, 20, 30, and 40 mm) (Fig. 3e, f), indicating that the mechanism of soft demoulding is significantly different from the rigid demoulding shown previously (see "Methods" section 'Mechanical characterization and Demoulding tests'). As shown in Supplementary Fig. 4a, b, when subjected to a stretching force, the strain stably increases, and the peel angle expends, while the force first surges under a short strain and then remains stable over a large range for the simulation results (see "Methods" section 'Simulation of the deformed angle'). It is observed in our soft demoulding experiments that the onset of peeling occurs when the peel force reaches a plateau (Supplementary Fig. 4c). For larger diameters, the peel force increases, and the consequent peel angles are different

(Fig. 3d and Supplementary Fig. 4c). Demoulding failure always occurs in the situation where the template reaches its maximum strain before demoulding is initiated. According to the simulation, for the soft template (thermoplastic resin) in this work, demoulding fails when the diameter is less than 15.1 μm, as shown in Fig. 3d since the peel angle exceeds its largest value (30°). Moreover, according to different mechanical behaviours of soft templates, we built the demoulding model for the TPU filament (see Supplementary Note 1 and Supplementary Fig. 4d–h).

Herein, the magnitude of the pull-out force for the soft template is drastically smaller than that for the rigid template, according to peeling theory (Fig. 3f and Supplementary Fig. 4), since the direct pulling of the rigid template can be regarded as the peeling of zero angles. Hence the soft template is less susceptible to fracture and appliable for thin and high-aspect-ratio microchannels generation.

With soft demoulding, we fabricated microchannels with a diameter of as small as 10 μm (Fig. 4a). As shown in Supplementary Fig. 5a, b, a microchannel with an aspect ratio as high as 6000 (around 10 times higher than the previously available maximum value, 629[9]) was also generated. Moreover, the resultant microchannel inner surface is smooth ($S_a = 0.018$ μm) (Supplementary Fig. 2b), which benefits soft robotics, fluidic interactions, and optical applications, such as improving the burst pressure and cycle life of soft actuators[41],

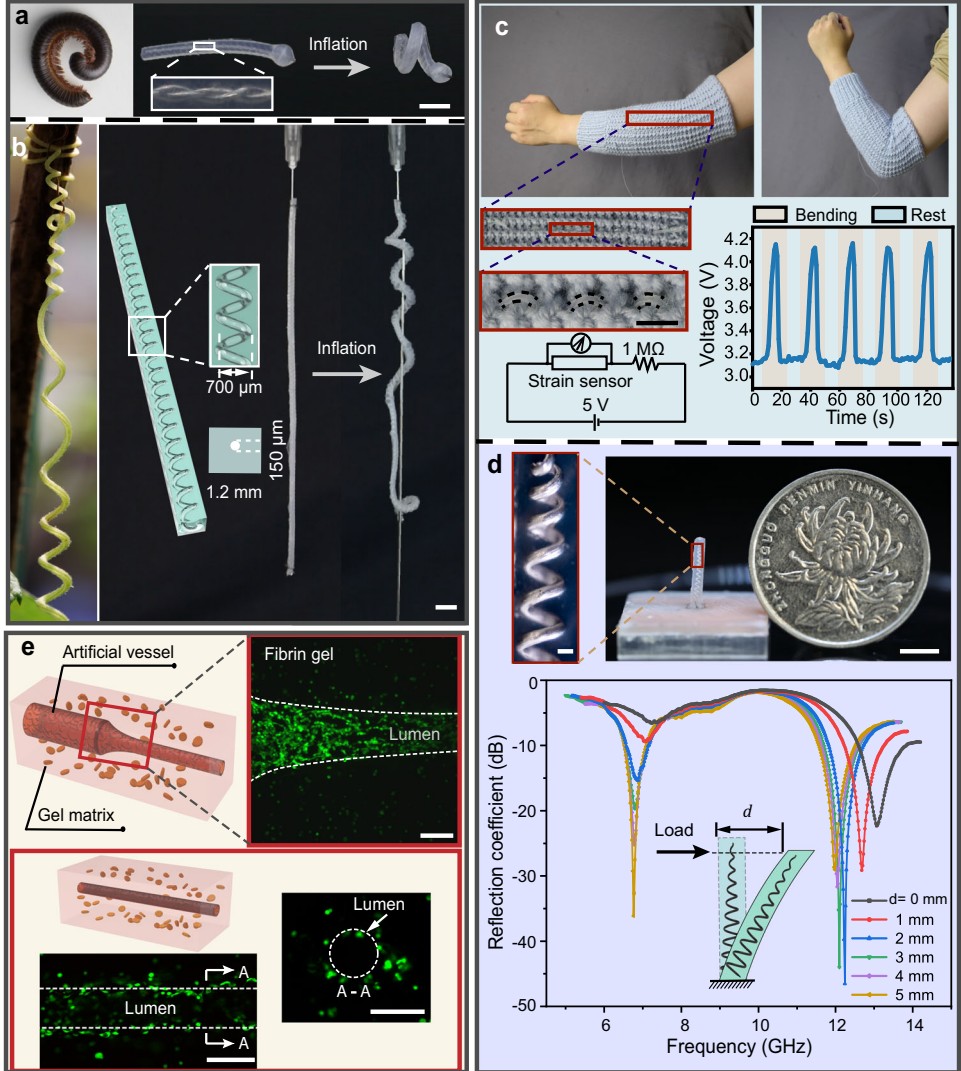

**Fig. 5 | Demonstrative applications of soft demoulding. a** The pneumatic soft robot in a twisting state mimics the millipede in a defence state (left inset) using the plectoneme microchannel. Scale bars **a**–**d**, 5 mm. **b** The long, soft tendril robot (length: 10 cm) containing a helical microchannel (diameter: 150 μm) (aspect ratio > 1600) climbs on a rod after being inflated, like the real tendril (left inset). **c** The soft, thin, long strain sensor (channel diameter: 150 μm, length: 15 cm) capable of acquiring the elbow motion. **d** The soft antenna containing a 3D helical microchannel (diameter: 180 μm) exhibiting different reflection coefficients under different deflection *d*. Scale bar (inset): 200 μm. **e** The artificial blood vessels in fibrin gels with HUVECs seeded, fabricated by soft demoulding. The confocal image of the cross-sectional views of the image (z-projection of a 250 μm stack) of the tapered artificial vessel (the minimum diameter: 250 μm, the maximum diameter: 500 μm) and the straight artificial vessel (diameter: 150 μm) after one day of HUVECs seeding. The confocal images of the fibrin gel after 1–2 days of culture stained with live (green)/dead (red) essay. Images in **e** are representative of three independent artificial vessels (experimental replicates). Scale bars, 200 μm.

enhancing the switching effect of microfluidic valves[42], and reducing the optical intensity loss for optical waveguides[43] since the template is deformable and the radial dimensions decrease when it is extracted, which largely reduces the pull-out force. Moreover, we fabricated microchannels of various shapes from 1D to 3D patterns, including a taper, a helix, a saddle, and a tree-like structure (Fig. 4b–q and Supplementary Fig. 5). Moreover, the demoulding possibility for the spindle-shaped soft template was discussed in Supplementary Note 2. Compared to other methods, soft demoulding can generate both higher geometric flexibility and smaller feature size[44] (the two primary features of microchannels) (Fig. 4r and Supplementary Table 1), comparable with human capillaries in terms of complexity and dimension.

### Applications for soft demoulding

Here, we demonstrate the extensive applications and significant impact of soft demoulding in soft robotics, wearable sensors, soft antennas, and artificial vessels (Fig. 5). Miniature soft robots have attracted increasing interest in recent decades due to their excellent compliance and adaptivity[8,45] in minimally invasive surgery, inspection, and search and rescue[35,46]. The fabrication of micro-chambers is challenging, particularly when the characteristic dimension is small and the topology is complex. For example, directly pulling out the template from the matrix can only create a simple straight chamber. Emerging solutions, such as chemical crosslink bonding interfaces[47], photo-curing 3D printing in monolithic structures[48], and dip-coating methods[7], suffer from rough surfaces, weaker strength, high time consumption, or limited shapes and sizes.

Herein, our soft demoulding method can create miniature soft robots with smooth, complex microchannels and monolithic structures. For example, inspired by worms that can curl up for defence, a miniature worm robot capable of bending at an angle larger than 450° was created (Fig. 5a, Supplementary Movie 2, and "Methods" section 'Fabrication of the soft worm robot'), which is applicable to delicate manipulation and grasping during surgeries[49]. Its inner chamber with a

plectoneme structure (diameter: 200 µm, see Supplementary Fig. 6a) fabricated by the soft demoulding method possesses a series of concaves in the elastic matrix, enabling a more effortless and quicker deformation of the actuator under inflation than a chamber with a constant cross section[50]. We also demonstrated a tendril-inspired ultra-long, soft robot (Fig. 5b and Supplementary Fig. 6b, and Supplementary Movie 3) integrating a helical microchannel (diameter: 150 µm) with an aspect ratio of more than 1600 (see Methods section Fabrication of the soft tendril robot). Pressured by air, the soft tendril robot winded around a rod accordingly, which can be applied for fixation and grasping, such as fixing and monitoring nerval activity[51], by imitating the tendril's survival strategy. These prototypes reveal the promising potential of creating more complex channels for more versatile miniature soft robots based on soft demoulding.

Furthermore, using the soft demoulding technique, we fabricated a stretchable sensor with smooth and round cross-sectional ultrathin channels (Supplementary Fig. 7a and "Methods" section 'Fabrication of the soft wearable strain sensor'). The emerging stretchable elastomeric sensors, although promising for wearable devices and human-machine interfaces[10,15,52], may cause uncomfortable and restricted during wearing due to the bulky structures. These problems mainly result from the limited fabrication methods. With soft demoulding, circular channels can be easily fabricated and can generate a more predictable resistive change response during extension (see Supplementary Note 3, Supplementary Fig. 7b–e, and Supplementary Movie 4) due to the isotropic cross section. We demonstrated a cotton-thread-like ultralong strain sensor fabricated by this method and seamlessly integrated it into a knitted sleeve. The soft strain sensor with the circular microchannel was manually sewn into a woven sleeve (Fig. 5c). The sensor was compatible with the sleeve and almost invisible because of its thin, long, and semitransparent characteristics, which are essential for wearable devices. This sensor accurately measured the elbow motion by the voltage signal variation (Fig. 5c and Supplementary Movie 5).

Moreover, a soft and mechanically tunable micro-antenna with a 3D helical conductive structure was fabricated by soft demoulding (see "Methods" section 'Fabrication of the soft antenna'). Existing small antennas, although critical for wearable devices, human-machine communication systems, and implant devices[53], are constricted by simple structures, such as rodlike or planar geometries[54,55], or supports for the 3D structures[55,56]. In addition, the large size (several centimetres) and rigid frame of current antennas hinder them from more extensive applications. Here, we fabricated a soft micro-antenna (10 mm × 1.2 mm × 1.2 mm) containing a 3D helical microchannel (channel diameter: 180 µm, helical structure diameter: 450 µm at the minimum and 900 µm at the maximum) infused with the liquid metal (Fig. 5d and Supplementary Fig. 8), while most previous small antennas were in the centimetre scale[53,56]. The 3D conducting structure offers a more compact dimension for confined environments. Moreover, with the 3D helical structure, the antenna presents low reflection coefficients of −6.6 and −22.3 dB (lower than −10 dB is sufficient for commercial antennas[57]) at two resonant frequencies (6.8 and 13.1 GHz), respectively. Being bent, the micro-antenna is mechanically tunable in two broad resonant frequency ranges (from 6.8 to 7.3 GHz and 11.9 to 13.1 GHz), and the reflection coefficient becomes lower for better signal transmission, as shown in Fig. 5d (see "Methods" section 'Reflection coefficient test for the soft antenna'). Therefore, the fabrication method of soft demoulding for micro-antenna provides a new approach toward compact soft wireless electronics.

Finally, using the soft demoulding technique, we created microchannels in the agarose gels and artificial blood vessels with straight and tapered structures. Microchannels are essential for, e.g., 3D tissue engineering development, disease analysis, and drug discovery[3,4,12]. Within the channels, cells can live longer and are able to develop into an organoid with a large volume, which is more suitable for customized medicine screening[4]. The formation of round cross sections and complex geometries, although critical for resembling the rheologic properties of blood flow[58] and the vascularization of a large tissue[5], is challenging. Previous rigid template demoulding is inapposite for a fragile matrix, and in the template dissolution method, the residual solvent and contaminant are cytotoxic to living cells[18,23]. Therefore, our solvent-free and soft demoulding process requiring gentle force is a superior approach for cell culturing in vitro. Here, we first fabricated a microchannel with a spindle-shaped structure and a microchannel with a narrow neck in agarose gels (Supplementary Fig. 9a, b, and Methods section Fabrication of biocompatible microchannel structures), which can be used for vascular disease models. Furthermore, we built a round cross-sectional artificial vessel (diameter: 150 µm) and a tapered one (the minimum diameter: 250 µm, and the maximum diameter: 500 µm) by seeding human umbilical vein endothelial cells (HUVECs) into the microchannel within the fibrin gel matrix (see Fig. 5e, Supplementary Fig. 9c–f, and "Methods" section 'Fabrication of the artificial vascular model'). After culturing for 1–2 days, the microchannel remained circular, and cells survived around the microchannel since nutrition can penetrate the porous structure of the artificial vascular wall. The cells near the channel exhibited a high survival rate (Fig. 5e), demonstrating that the channel architectures can provide functional nutrient transport to the near cell matrix. Moreover, this vascular structure can be used for simulating vessel growth and further vascular disease prediction. To exhibit the advantages of soft demoulding, we employed two rigid templates for fabricating microchannels in fragile fibrin gels, but the large shear force caused the microchannels to rupture (see Supplementary Fig. 10a, and "Methods" section 'Fabrication of microchannels in fibrin gels by rigid demoulding'). We also verified the negative effect of acetone, which is employed in the matrix by matrix swollen[16,22] and template dissolution methods[18,25], on cell growth in artificial vessels. When introducing acetone to the gels, the death rate of the 3T3 cells increased accordingly, as shown in Supplementary Fig. 10b. Therefore, with the gentle and solvent-free soft demoulding technology for 3D complex microchannel fabrication, a more complex artificial vascular system could be fabricated for prospective applications.

## Discussion

In summary, this paper presents a promising solution for the microchannel formation methodology based on soft templates and peeling-dominant removal procedures. Compared to traditional approaches, e.g., photolithography, this method is simple, fast, solvent-free, and can generate a microchannel with a super-large aspect ratio, smooth surface, and complex 3D geometries. We demonstrate its extensive applications by multiple prototypes, including a soft worm robot with a plectoneme-shaped structure, an ultra-long tendril robot containing a helical microchannel (diameter: 150 µm, aspect ratio: more than 1600), a thread-like biocompatible wearable sensor, a soft antenna containing 3D helical microchannel with variable diameter, and a thin artificial blood vessel in tapered geometry. Our soft demoulding method also offers a promising approach to the microfluidics field and tissue engineering.

Currently, the materials and fabrication method for soft templates employed in this work may limit it for more intricate and delicate microchannels generation. Future research will focus on the improvement of design (e.g., computer-assisted design[59]), fabrication process (e.g., employing high-precision moving stages for direct ink writing[34], and electrospinning technology[60]), and materials (e.g., hydrogels, which show ultra-large stretch, toughness, and self-lubrication properties[61]) for soft template generation and matrix formation to enhance the precision, complexity, versatility, and biocompatibility of the methodology for more extensive applications.

## Methods

### Fabrication of soft templates

First, in the thermal drawing process, we heated the thermoplastic resin (hot melt adhesive 3748Q, 3M) to 120 °C into its molten state and kept the molten state for 5–10 min. Then we cooled the material to 100 °C to reach its optimal drawing state. After that, we dipped the tip of a needle in the melt. The needle (diameter: 400 μm) was installed on a vertically moving stage. By controlling the moving stage speed and raising the needle, we could draw out a filament by the needle (Fig. 2a). In a similar way, thermoplastic polyurethane (TPU-95A, eSUN) was used for the tests.

Except for simple straight filaments, different template patterns (1D, 2D, or 3D) could be generated by altering the thermal drawing process and combining postprocessing and assembly (Fig. 2c–m). The taper-shaped soft template was fabricated by an accelerated drawing process since the filament diameter decreased with increasing speed (Fig. 2d). The template with a branched shape was drawn by two needles, moving in the same direction first and then in different directions (Fig. 2e). When the heating temperature increased from 100 to 130 °C, the polymer melt drawn needed a longer solidification time, and the melt fell like dripping water, forming a spindle-knotted structure on the filament (Fig. 2f) due to the synergistic interaction of the viscosity and surface tension[38]. To fabricate a helical structure, we stretched and released the filament for more than 10 cycles before the filament cooled down and then left it free to curl under residual stress (Fig. 2g). The plectoneme structure (Fig. 2h) was automatically formed by twisting the two ends of the straight filament along in inverse directions simultaneously.

For more complex 3D template constructs, such as a conical structure and a hyperboloid structure, we first fabricated a 3D frame and then bonded the templates on the frame with the corresponding pattern, as demonstrated in Fig. 2i, k. A PDMS saddle surface mould was first fabricated by mould injection, and then the soft template pattern in a saddle geometry was produced by attaching the soft filaments to the saddle mould (Fig. 2j). Grooves were designed on the saddle mould for template fixing. A tree-like soft template was created by gluing thin filaments as branches to a larger filament structure, as shown in Fig. 2l, m. All the soft templates were made of hot melt adhesive 3748Q. The images of the soft templates (Fig. 2c–h) were observed by a confocal laser scanning microscope (VK-X1000; Keyence).

### Fabrication of the microchannels by soft demoulding

With soft templates, microchannels can be generated by removing the templates from the matrix. As shown in Supplementary Fig. 1a soft template was first fixed at the centre of a mould. Next, the PDMS precursor (Sylgard 184, Dow Corning; 10: 1 weight ratio) was poured into the 3D printed mould to submerge the soft template and was thermally cured at 60 °C for 12 h. Then, the cured PDMS was separated from the mould. A force was applied to both ends of the soft template to stretch and peel the template out of the matrix. Finally, a PDMS matrix with a microchannel inside was produced. Microchannels generated via this soft demoulding technology are shown in Fig. 4a–q. Furthermore, these microchannels were illustrated by infusing fluorescent dye and imaged with a confocal microscope (Nikon A1, Nikon), as shown in Supplementary Fig. 5. The 10 μm microchannel (Fig. 4a) was fabricated by the TPU filament. The other microchannel structures shown in Fig. 4 and Supplementary Fig. 5 were fabricated by the hot melt adhesive 3748Q. The images of the microchannels (Fig. 4a–k) were observed by a confocal laser scanning microscope (VK-X1000; Keyence).

### Mechanical characterization

All stress-strain tests for the rigid and soft templates were conducted by a universal tensile tester (C42. 203, MTS) at room temperature. Four

different filiform samples, the thermoplastic resin (hot melt adhesive 3748Q, 3M), thermoplastic polyurethane (TPU-95A, eSUN), copper wires (Dupont line 40P, Risyn), and nylon fibre (No. 0.4, YNKOO), were subjected to the clamps of the test machine respectively, with a 10 mm/min stretch rate.

### Demoulding tests

All demoulding tests were conducted by a universal tensile tester (C42. 203, MTS) at room temperature. The geometry of the sample is shown in the inset of Fig. 3e. The width and depth of the elastic matrix of samples are 10 and 1 mm, respectively. The matrix material is PDMS. Each sample remained a 10 mm unembedded length for clamping during the tests. For rigid demoulding, four different embedded lengths (5, 10, 20, and 30 mm) of copper and nylon filaments were prepared. For soft demoulding, four different embedded lengths (10, 20, 30, and 40 mm) of thermoplastic resin and TPU filaments were prepared. All the samples were tested with a 60 mm/min stretch rate. Demoulding data were analyzed by Origin 2018.

### Simulation of the deformed angle

A finite element analysis (FEA) model (ABAQUS Explicit 2020) was built to estimate the peel angle since this angle is difficult to measure by tests. The materials were set as thermoplastic resin and hyper-elastic polyurethane with a diameter of 400 μm and a length of 1.2 mm. First-order cubic C3D8 elements with normal integration were used in the model. The angle θ is defined as the edge deformation of the first unit from the fixed side, as shown in Supplementary Fig. 4a.

### Fabrication of the soft worm robot

The soft worm robot was fabricated using silicone (E610, Shenzhen Hong Ye Jie Technology Co.) with a thermoplastic resin template (hot melt adhesive 3748Q). The robot body was fabricated following the same protocol as shown in Supplementary Fig. 1. The plectoneme microchannel had a slight slant (approximately 2.7°), as shown in Supplementary Fig. 6a. Due to the slant angle, an out-of-plane force and bending were generated under inflation. Additionally, the isosceles trapezoid geometry design in the cross section of this robot enlarged the bending motion (see Supplementary Fig. 6a) by differing the bending stiffness along the two sides.

### Fabrication of the soft tendril robot

The soft tendril robot was created by first embedding a long helical TPU template in the silicone matrix (Ecoflex 0050, Smooth-on) and peeling the template out. A microchannel 150 μm in diameter and 25 cm in length (see Supplementary Fig. 6b) was produced, corresponding to a super-high aspect ratio of 1600. Finally, a small amount of precursor was used to block the tip of the tendril robot. When the pressure was applied, this soft straight structure transformed into a winding state to mimic the climbing strategy of tendrils (see Fig. 5b).

### Fabrication of the soft wearable strain sensor

The microchannel for the strain sensor was fabricated by following the same protocol shown in Supplementary Fig. 1. First, a straight microchannel (length: 20 cm and diameter: 150 μm) was generated in the silicone matrix (Ecoflex 0050, Smooth-on) by a TPU filament. Then, saturated sodium chloride – glycerol was injected into the tubular structure to function as the conductor in the elastic sensor (Supplementary Fig. 7a). Finally, the channel was sealed by the silicone precursor. The voltage data were acquired by LabVIEW 2019. The simulation of microchannels containing different cross-sectional geometries under stretching was performed in ABAQUS Explicit 2020, and the data were analyzed by MATLAB 2019a.

## Fabrication of the soft antenna

The soft helical template was generated by fixing a straight TPU filament onto a metal cone and heating the filament with a heat gun (200 °C) to reach the various diameter helical soft template. Then, this soft helical template was first fixed to a mould, and the elastomer precursor (Ecoflex 0050, Smooth-on) was poured into the mould. After the precursor was cured, the microchannel was generated by pulling the template out. Next, a syringe injected the liquid metal (−19 °C, Dingguan Metal Technology Co.) into the microchannel. Finally, both ends of the microchannel were sealed by the silicone precursor.

## Reflection coefficient tests for the soft antenna

The soft antenna was fixed to a 3D-printed frame (Clear, Formlabs), and a copper foil was attached to the bottom of the frame working as the ground plate. Next, the antenna was fixed to the clamp, and the sliding state controlled the deflection, as shown in Supplementary Fig. 8c. Then a vector network analyzer (N5227B, Keysight) was connected to the antenna for the reelection coefficient test.

## Fabrication of biocompatible microchannel structures

With soft demoulding, which is solvent-free and gentle in force, we fabricated different biocompatible microchannels from an aqueous agarose solution (1.5% w/v). First, the soft filaments (Hot melt adhesive, 3748Q) were fixed to a 3D printed mould. Then, the liquid agarose was poured into the mould to immerse the templates. After the agarose solidified, the agarose matrix was separated from the mould. Finally, the soft templates were demoulded from the agarose gels, and a microchannel lattice was formed, as shown in Supplementary Fig. 9a, b.

## Fabrication of the artificial vascular model

Human umbilical vein endothelial cells (HUVECs) were purchased from Lonza (catalogue number: CC-2517); BALB/C 3T3 cell line was obtained from the Cell Resource Centre, Peking Union Medical College (resource number: 1101MOU-PUMC000186). No cell lines used in this study were listed in the database of known misidentified cell lines maintained by the International Cell Line Authentication Committee. For cell culture, the HUVECs were maintained in Endothelial Cell Medium (ECM, ScienCell) containing 5% (v/v) Fetal bovine serum (FBS, ScienCell), 1% (v/v) endothelial cell growth supplement (ECGS, ScienCell), and 1% (v/v) penicillin-streptomycin (ScienCell). BALB/c 3T3 cells were cultured in Dulbecco's Modified Eagle Medium (DMEM) with high glucose (4.5 g/L, Gibco) containing 10% FBS (Gibco) and 1% penicillin-streptomycin (Gibco). All cells were maintained at 37 °C with 5% $CO_2$ in a humidified incubator.

The mould for the artificial vessel was first immersed into 75% ethanol overnight and sterilized with ultraviolet (UV) light for one hour before use. Fibrin gel (21.5 mg/mL) was formed by dissolving fibrinogen in DMEM with high glucose (4.5 g/L, Gibco), 10% FBS, and 1% penicillin-streptomycin. The centrifugation ($1500$–$2000 \times g$) was applied to remove air bubbles. The 3T3 cell pellet was gently resuspended in the fibrinogen solution to a concentration of $5$–$10 \times 10^5$ cells/mL. As the cells were uniformly resuspended and mixed with fibrinogen, thrombin was added to a final concentration of 3 U/mL. The fibrin gel mixture was quickly added to the model and placed in a 37 °C incubator for crosslinking. After 6 h of incubation, the mould was visualized under a stereomicroscope in the biological safety cabinet, and the soft template (Hot melt adhesive, 3748Q) was demoulded from the gel to generate the microchannel. HUVECs suspension was concentrated to $6 \times 10^6$ cells/mL, and first seeded into the bottom of the channel. After one hour of static culturing, the device was turned over, and the HUVECs suspension was seeded into the top of the channel with another 2 h of incubation (Supplementary Fig. 9c). The non-adherent cells and cell debris within the microchannel were

removed by fresh medium. The mould was cultured with ECM containing 5% FBS, 1% ECGS, and 1% penicillin-streptomycin for 8 h under static conditions to allow the cells to adhere and spread before introducing hemodynamic flow by rocking platform (Supplementary Fig. 9d).

The fibrin gels with microchannels were stained with a fluorescent live/dead assay after 1–2 days of culture. Calcein acetoxymethyl (Calcein AM, "live", Yeasen Biotechnology) and propidium iodide (PI, "dead", Yeasen Biotechnology) were diluted to 5 and 1.5 mM as stock solutions, respectively. Calcein AM and PI were diluted with phosphate-buffered saline (PBS) to final concentrations of 8 and 3 μM and maintained with the gel for 20 min at 37 °C. The microchannel within the fibrin gel was gently washed with PBS three times and then imaged with a confocal microscope (Nikon A1, Nikon).

To verify the negative effect of the acetone concentration on the death rate of 3T3 cells, we treated the fibrin gels with acetone in different concentrations (0, 0.5, 1, and 2%, volume ratio) in the fibrin gels. The 3T3 cell concentration in the fibrin gels was $1.5 \times 10^6$ cell/mL. The artificial vessels were stained with a fluorescent live/dead assay after 2 days of culture, as shown in Supplementary Fig. 10b.

## Fabrication of microchannels in fibrin gels by rigid demoulding

We first prepared two rigid templates, a nylon filament (diameter: 200 μm) and a Nitinol filament (diameter: 300 μm), for rigid demoulding. The microchannels in the fibrin gels were fabricated by following the same protocol shown in Supplementary Fig. 1. The 3T3 cell concentration in the fibrin gels was $5 \times 10^5$ cell/mL, and the fabricated microchannels are shown in Supplementary Fig. 10a.

## Reporting summary

Further information on research design is available in the Nature Research Reporting Summary linked to this article.

# Data availability

The data that support the findings of this study are available within the paper and its Supplementary Information and from the corresponding author upon request.

# Code availability

The code in this study has been deposited in the Code Ocean repository (https://doi.org/10.24433/CO.0908662.v1) or can be requested from the corresponding authors.

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

## Acknowledgements

We acknowledge the support of the following: National Natural Science Foundation for Young Scientists of China (51905256, H.W.), Natural Science Foundation of Guangdong Province of China (2020A1515010955, H.W.), Science, Technology and Innovation Commission of Shenzhen Municipality (ZDSYS20200811143601004, H.W.), and Natural Science Foundation of Liaoning Province of China (State Key Laboratory of Robotics joint funding, 2021-KF-22-11, H.W.), and Southern Marine Science and Engineering Guangdong Laboratory (Guangzhou), (K19313901, H.W.), and National Natural Science Foundation of China (31970752, P.Q.), Science, Technology, Innovation Commission of Shenzhen Municipality (JCYJ20190809180003689, JSGG20200225150707332, JSGG20191129110812708, P.Q.). The authors acknowledge the assistance of SUSTech Core Research Facilities.

## Author contributions

D.F. and H.W. conceived the concept and designed the research. D.F., Xi Yuan, W.W., R.Z., Xin Yang, Y.M., and C.L. conducted the experiments. W.W. and Y.L. contributed to the simulation. D.F., W.W., and Y.L. performed data analysis. D.F., Xi Yuan, C.C., H.W., and P.Q. completed the manuscript. H.W. and P.Q. supervised the study. All authors provided feedback.

## Competing interests

H.W. is applying for one patent related to the described work. The other authors declare no competing interests.
