## [Peer Review File · Nature Communications]

Self-shrinking soft demoulding for complex high-aspect-ratio microchannelsReviewers' Comments:

Reviewer #1:

Remarks to the Author:

The manuscript describes a new soft-lithographic fabrication technique for microscale channels and its applications. The technique is clearly novel and seems significant for a wide range of applications. The manuscript is well-written with adequate levels of clarity and visual impact. The significance of the work and the way it is presented, however, doesn't seem to be sufficient for publication in Nature Communications. A major revision is required. The reviewer's comments are provided below to help the authors improve their work and manuscript.

(1) The work's significance can be presented best by demonstrating an application that can exclusively be realized by the new technique. The current work is weak in that respect because most exemplary applications, both in main text or supplementary information, can be realized by existing techniques. The author's new technique really shines when it is utilized to make complex, highly curved networks of really thin, axially modulated (such as twisted or tapered) microchannels. A few results shown in Fig. 3, esp. those in Fig. 3j~l, fit the description but most others don't. In particular, the big channels used for the flower actuator (Fig. 4c), tactile sensor (Fig. 4e), and artificial blood vessel (Fig. 4f) are totally against the purpose of the author's new technique. The sub-mm-scale straight channel network in agarose gel is actually an anti-example because it can be easily realized by hard templates such as glass fibers. The reviewer suggests these non-essential applications be replaced by more critical ones which fully exploits the strength of the proposed technique.

(2) The way the work is presented also needs improvements. In the figures, the principle, fabrication process, results, and applications are mixed up. They need to be separated. Many results in Extended Data Fig. 3 seems essential to describe the significance of the work. They should be brought back to the main text.

(3) The claim of "snake shedding inspired" seems not firmly supported in the manuscript. It is a nice way of explaining the process. However, there is no theoretical or experimental support to justify the claim. There is no citation of references to support the claim either. The authors are suggested to back up the claim by elaboration and/or references or to remove the claim. The work is significant by its own right, without the need for a "nature-inspired" tag.

(4) The description of the work (from page 12) also requires further clarifications.

(a) It is often not clear whether the authors used 3748Q or TPU-95A. Please specify the material used for each experiment.

(b) In page 13, comparisons were made to emphasize the advantage of the present work. First of all, this should be a part of the main text, not the Method. Second, this work is strongly influenced by the work by Verma et al. So, comparison with the use of nylon templates seems informative, and even imperative.

(c) In Line 396 (page 15), the natural question is "What will happen to thermoplastic urethane if it were subjected to this modeling?" Please address this issue in the revision.

(d) The "bulge" (Fig. 2f, for example) can invalidate the "shedding" process described in Fig. 1. So, the reviewer anticipates a certain size limit in the bulge which restricts the utilization of the proposed technique. Please comment on this possibility.

Overall, the manuscript describes a new and potentially useful microfabrication technique but its presentation and application need to be greatly improved for publication in Nature Communications.

Some miscellaneous points include:

(a) The sentence in Line 73~74 is repeated and redundant.

(b) In Line 93, "Except for" seems to mean "In addition to".

- (c) Figure 3u seems too busy. Just use the last names.
(d) Extended Data Fig. 4c doesn't seem to play any role.

Reviewer #2:

Remarks to the Author:

In this paper, the authors report a method for creating microfluidic channels using soft demolding, inspired by snake shedding. The soft demolding allowed them to create fairly complex, circular channel structures, with aspect ratios well above conventional values (>6000). As a result, the authors fabricated 3D microfluidic channels with flower-like opening and closing functions, microfluidic channels with skin-like sensor systems, and microfluidic channels with blood vessel models. In my opinion, the results of this paper are of interest to the broader community of microfluidics researchers working in different areas and are likely to be of great interest to the readers of Nature Communications. Therefore, in my opinion, this paper is worthy of publication, but it would be desirable to make some revisions in the points listed below.

1. I like how the authors consider ideas about the microfluidic fabrication method by soft demolding. However, I miss a control experiment to show that the vascular model obtained in their experiment cannot be obtained with a channel produced by soft lithography. In other words, what they have shown in Fig. 4f is reproduced by soft lithography without their technique. I think this is important to support their claim that "using soft lithography simply yields rectangular 2D channels and does not allow for ultra-thin, 3D complex structures".
2. Specific reference to the need for complex 3D and high aspect ratio microfluidic channels is needed. The sentence of "Most of the current fabrication methods are inadequate for biological applications that strictly require nontoxic and biocompatible elements" is unclear as to the reason for the need for complex 3D and high aspect ratio microfluidic channels.
3. Need a table summarizing soft lithography, emerging methods and soft demolding with respect to 3D complexity, aspect ratio, channel geometry, toxicity, etc.
4. Interesting results on the miniaturization and complexity of a single structure are presented, but the question remains on the complexity of the aggregate. The current state of the art is shown in microchannel fabrication in Fig. 3u, but it seems to be due to the lack of explanation on the difficulty of assembling these structures, although the shape at the level of a single structure is mentioned.
5. The merits of this paper are (1) a high aspect ratio and (2) the ability to fabricate structures of small size, and the results of each were shown. There was no mention, however, of (1) and (2) being necessary at the same time, and we felt the need to mention this.
6. In the case of fabricating assembled structures in combination with existing techniques, I felt that the size of the structures is several hundred μm , which obscures the advantages of their results. In addition, it is necessary to follow up on the mechanical resistance of the mold in order to expand the advantages of this method to "high aspect ratio" and/or "ultra-thin" microchannels.

Reviewer #3:

Remarks to the Author:

This paper by Fan et al. reports a fabrication method of microchannels. The authors say this method was inspired by snake-shedding, but I don't think this method is superior to other methods. Although it may be unique as a fabrication method, the structure fabricated in the authors' demonstration is not very useful. Fabricating simple microchannel (such as straight) is easy, but fabricating complex channel structures can be tedious to assemble soft templates. Moreover, accurate placement is not easy. I think any of the structures they have shown can be made without using this method. Therefore, I concluded that this paper has not reached the scientific level worthy of being published in Nature Communications.

Specific comments:

1. With the electrospinning method, smaller diameter products can be produced. Using it as a template, we can create a microchannel by using the melting point difference. Also, I think that a similar microchannel can be fabricated by 3D laser processing. The surface roughness of the inner surface in the microchannel may be superior to our method, but I don't know if there is any application that can take advantage of it.
2. Line 24 and line 124: Normally, the aspect ratio of the microchannel is expressed by width-to-depth, but this 6000 is width-to-length. Since it is circular, the aspect ratio is only 1. Adopting your definition, there are many hollow tubes with aspect ratios over 6000 (eg capillary tubes).
3. Line 85: The authors say "providing great design flexibility", but I don't think it's so flexible because it can only be fabricated by combining soft templates.
4. There are many references where the journal name is not capitalized.
5. Reference 20: Kobayashi, S. is correct.
6. Reference 25: Sensors and Actuators A, 226, 137-142 (2015).

Reviewer #1(Remarks to the Author):

Paper summary: The manuscript describes a new soft-lithographic fabrication technique for microscale channels and its applications. The technique is clearly novel and seems significant for a wide range of applications. The manuscript is well-written with adequate levels of clarity and visual impact. The significance of the work and the way it is presented, however, doesn't seem to be sufficient for publication in Nature Communications. A major revision is required. The reviewer's comments are provided below to help the authors improve their work and manuscript.

RC: (1) The work's significance can be presented best by demonstrating an application that can exclusively be realized by the new technique. The current work is weak in that respect because most exemplary applications, both in main text or supplementary information, can be realized by existing techniques. The author's new technique really shines when it is utilized to make complex, highly curved networks of really thin, axially modulated (such as twisted or tapered) microchannels. A few results shown in Fig. 3, esp. those in Fig. 3j~l, fit the description but most others don't. In particular, the big channels used for the flower actuator (Fig. 4c), tactile sensor (Fig. 4e), and artificial blood vessel (Fig. 4f) are totally against the purpose of the author's new technique. The sub-mm-scale straight channel network in agarose gel is actually an anti-example because it can be easily realized by hard templates such as glass fibers. The reviewer suggests these non-essential applications be replaced by more critical ones which fully exploits the strength of the proposed technique.

AR: The authors thank the reviewer's valuable suggestions. The advantage of our soft demoulding technology is fabricating microchannels with high aspect ratios and 3D complex geometries. Initially, the demonstrational works containing big channels shown in Fig. 4 and Extended Data Fig. 7 (in the last version) mainly showed the 3D complex features made by this technique, and the large aspect ratio was displayed by the soft tendrill robot.

According to the reviewer's suggestions, we have created new samples, including a soft tendrill robot containing a 3D helical microchannel (diameter: 150 μm) of a high aspect ratio (over 1600), a soft antenna containing a 3D helical microchannel (diameter: 180 μm), microchannels with spindle-shaped and narrow-necked structures in agarose gels, a thin straight artificial vessel (diameter: 150 μm), and a tapered artificial vessel (the minimum diameter: 250 μm , the maximum diameter: 500 μm).

We have added the above information in detail in the main text and Fig. 5, Supplementary Fig. 6, 8, and 9.

We revised the 1st paragraph of "Abstract" section as follows:

"We demonstrate the vast applicability and significant impact of this technology in multiple scenarios, such as a tendrill robot capable of winding by a super-long helical microchannel, a soft mechanically-tunable miniature antenna with a 3D helical microchannel, and artificial

vessels with straight and tapered structures capable of transporting nutrition to the surrounding cells."

We also revised the description in the 8th paragraph of "Results" section:

"We also demonstrated a tendril-inspired ultra-long, soft robot (Fig. 5b, Supplementary Fig. 6b, and Supplementary Movie 3) integrating a helical microchannel (diameter: 150 μm) with an aspect ratio of more than 1600 (see "Methods" section 'Fabrication of the soft tendril robot'). Pressured by air, the soft tendril robot winded around a rod accordingly, which can be applied for fixation and grasping, such as fixing and monitoring nerval activity⁵¹, by imitating the tendril's survival strategy."

In the 10th paragraph of "Results" section:

"Moreover, a soft and mechanically tunable micro-antenna with a 3D helical conductive structure was fabricated by soft demoulding (see "Methods" section 'Fabrication of the soft antenna'). Existing small antennas, although critical for wearable devices, human-machine communication systems, and implant devices⁵³, are constricted by simple structures, such as rodlike or planar geometries^{54, 55}, or supports for the 3D structures^{55, 56}. In addition, the large size (several centimeters) and rigid frame of current antennas hinder them from more extensive applications. Here, we fabricated a soft micro-antenna (10 mm x 1.2 mm x 1.2 mm) containing a 3D helical microchannel (channel diameter: 180 μm , helical structure diameter: 450 μm at the minimum and 900 μm at the maximum) infused with the liquid metal (Fig. 5d and Supplementary Fig. 8), while most previous small antennas were in the centimeter scale^{53, 56}. The 3D conducting structure offers a more compact dimension for confined environments. Moreover, with the 3D helical structure, the antenna presents low reflection coefficients of -6.6 and -22.3 dB (lower than -10 dB is sufficient for commercial antennas⁵⁷) at two resonant frequencies (6.8 GHz and 13.1 GHz), respectively. Being bent, the micro-antenna is mechanically tunable in two broad resonant frequency ranges (from 6.8 GHz to 7.3 GHz and 11.9 to 13.1 GHz), and the reflection coefficient becomes lower for better signal transmission, as shown in Fig. 5d (see "Methods" section 'Reflection coefficient test for the soft antenna'). Therefore, the fabrication method of soft demoulding for micro-antenna provides a new approach toward compact soft wireless electronics."

In the 11th paragraph of "Results" section:

"Here, we first fabricated a microchannel with a spindle-shaped structure and a microchannel with a narrow neck in agarose gels (Supplementary Fig. 9 a, b, and "Methods" section 'Fabrication of biocompatible microchannel structures'), which can be used for vascular disease models. Furthermore, we built a round cross-sectional artificial vessel (diameter: 150 μm) and a tapered one (the minimum diameter: 250 μm , and the maximum diameter: 500 μm) by seeding human umbilical vein endothelial cells (HUVECs) into the microchannel within the fibrin gel

matrix (see Fig. 5e, Supplementary Fig. 9 c-f, and "Methods" section 'Fabrication of the artificial vascular model')."

In the 12th and 13th paragraphs of "Results" section:

"We demonstrate its extensive applications by multiple prototypes, including a soft worm robot with a plectoneme-shaped structure, an ultra-long tendril robot containing a helical microchannel (diameter: 150 μm , aspect ratio: more than 1600), a thread-like biocompatible wearable sensor, a soft antenna containing 3D helical microchannel with variable diameter, and a thin artificial blood vessel in tapered geometry. Our soft demoulding method also offers a promising approach to the microfluidics field and tissue engineering.

Although our soft demoulding method shows advantages for fabricating 3D complex microchannels with a high aspect ratio, the materials and fabrication method for soft templates employed in this work may limit it for more intricate and delicate microchannels generation. Future research will focus on the improvement of design (e.g., computer-assisted design⁵⁹), fabrication process (e.g., employing high-precision moving stages for direct ink writing³⁴, and electrospinning technology⁶⁰), and materials (e.g., hydrogels, which show ultra-large stretch, toughness, and self-lubrication properties⁶¹) for soft template generation and matrix formation to enhance the precision, complexity, versatility, and biocompatibility of the methodology for more extensive applications."

In the 11th, 13th, and 15th paragraphs of "Methods" section:

"Fabrication of the soft antenna. *The soft helical template was generated by fixing a straight TPU filament onto a metal cone and heating the filament with a heat gun (200 °C) to reach the various diameter helical soft template. Then, this soft helical template was first fixed to a mould, and the elastomer precursor (Ecoflex 0050, Smooth-on) was poured into the mould. After the precursor was cured, the microchannel was generated by pulling the template out. Next, a syringe injected the liquid metal (-19°C, Dingshan Metal Technology Co.) into the microchannel. Finally, both ends of the microchannel were sealed by the silicone precursor."*

"Fabrication of biocompatible microchannel structures. *With soft demoulding, which is solvent-free and gentle in force, we fabricated different biocompatible microchannels from an aqueous agarose solution (1.5% w/v). First, the soft filaments (Hot melt adhesive, 3748Q) were fixed to a designed 3D printed mould. Then, the liquid agarose was poured into the mould to immerse the templates. After the agarose solidified, the agarose matrix was separated from the mould. Finally, the soft templates were demoulded from the agarose gels, and a microchannel lattice was formed, as shown in Supplementary Fig. 9 a and b."*

"The mould for the artificial vessel was first immersed into 75% ethanol overnight and sterilized with ultraviolet (UV) light for one hour before use. Fibrin gel (21.5 mg/mL) was formed by dissolving fibrinogen in DMEM with high glucose (4.5 g/L, Gibco), 10% FBS, and 1% penicillin-streptomycin. The centrifugation (1500-2000g) was applied to remove air bubbles. The 3T3 cell

pellet was gently resuspended in the fibrinogen solution to a concentration of $5-10 \times 10^5$ cells/mL. As the cells were uniformly mixed with fibrinogen, thrombin was added to a final concentration of 3 U/mL. The HUVEC suspension was concentrated to 1.5×10^6 cells/mL and seeded into the channels. The fibrin gel mixture was quickly added to the model and placed in an incubator for crosslinking. After 6 hours of incubation, the soft template (Hot melt adhesive, 3748Q) was demoulded from the gel to generate the microchannel. HUVECs suspension was concentrated to 6×10^6 cells/mL, and first seeded into the bottom of the channel. After one hour of static culturing, the device was turned over, and the HUVECs suspension was seeded into the top of the channel with another 2 hours of incubation (Supplementary Fig. 9c). The non-adherent cells and other cell debris within the microchannel were removed by fresh medium. The mould was cultured with ECM for 8 hours under static conditions to allow the cells to adhere and spread before introducing hemodynamic flow by rocking platform (Supplementary Fig. 9d)."

Fig. 5 Demonstrative applications of soft demoulding. **a**, The pneumatic soft robot in a twisting state mimics the millipede in a defence state (left inset) using the plectoneme microchannel. Scale bars (a-d), 5 mm. **b**, The long, soft tendrill robot (length: 10 cm) containing a helical microchannel (diameter: 150 μm) (aspect ratio >1600) climbs on a rod after being inflated, like the real tendrill (left inset). **c**, The soft, thin, long strain sensor (channel diameter: 150 μm , length: 15 cm) capable of acquiring the elbow motion. **d**, The soft antenna containing a 3D helical microchannel (diameter: 180 μm) exhibiting different reflection coefficients under different deflection. Scale bar (inset): 200 μm . **e**, The artificial blood vessels in the fibrin gel with HUVECs seeded. The confocal image of the cross-sectional views of the image (z-projection of a 250 μm stack) of the tapered artificial vessel (the minimum diameter: 250 μm , the maximum diameter: 500 μm) and the straight artificial vessel (diameter: 150 μm) after one day of HUVECs seeding. The confocal images of the fibrin gel after 1-2 days of culture stained with live (green)/dead (red) assay. Scale bars, 200 μm .

Supplementary Fig. 6 Applications of soft demoulding on soft robotics for shape morphing. **a**, The structure of the soft worm robot. Scale bar, 500 μm . **b**, The structure of the soft tendril robot and its winding state after inflation. The left inset shows the structure of the helical microchannel (scale bar: 200 μm). Scale bar, 5 mm.

Supplementary Fig. 8 Applications of the soft antenna. **a**, Fabrication process of the soft antenna with a 3D helical microchannel. A 3D helical soft filament is fabricated by first aligning a straight soft filament to a metal cone into 3D geometry and fixing the helical geometry by heat treatment. Then, the soft template is immersed into the liquid precursor and thermally cured. Next, the template is demoulded from the elastic matrix. Finally, the liquid metal is injected into the helical microchannel. **b**, The setup for the reflection coefficient tests.

Supplementary Fig. 9 Applications on biomedical devices based on soft demoulding methodology. **a**, The spindle-shaped microchannel in the agarose gel (left diameter: 100 μm , the maximum diameter of the spindle shape: 450 μm , and right diameter: 200 μm). **b**, The straight microchannel with a narrow neck (diameter: 300 μm , and the narrow diameter: 150 μm). **e**, The brightfield image of the straight artificial vessels (diameter: 150 μm) in the fibrin gels formed by soft demoulding. **f**, The brightfield images of the top view and cross-sectional view of the tapered artificial vessels (the minimum diameter: 250 μm , the maximum diameter: 500 μm). Scale bars, 200 μm .

RC: (2) The way the work is presented also needs improvements. In the figures, the principle, fabrication process, results, and applications are mixed up. They need to be separated. Many results in Supplementary Fig. 3 seems essential to describe the significance of the work. They should be brought back to the main text.

AR: The authors appreciate the reviewer's suggestions. According to the comments, we moved the previous subfigures b-d in Supplementary Fig. 3 into Fig. 3, and separated previous Fig. 3 into two figures to show the soft demoulding mechanism and microchannel structures, respectively, as follows:

Fig. 3 Mechanisms of soft demoulding and rigid demoulding. **a**, The stress-strain curves of a rigid template (copper) and a soft template (thermoplastic resin). **b**, The critical fracture force, and shear force of the rigid template (copper wire with 20 mm embedded length) for different filament diameters. The shadow region represents the range of the demoulding failure. **c**, The force/diameter – displacement curve with different embedded lengths for rigid templates (copper). The diameter of the copper wire is 80 μm . **d**, The critical fracture force and peel force varying with diameter of soft template (thermoplastic resin). The intersections in the plot indicate the minimum diameter for initiating the peel process with certain peel angles (0, 20, and 30 degrees). The minimum microchannel generated by this filament is limited by the largest peel angle (30 degrees). The shadow region represents the demoulding failure. **e**, The pull-out force of the soft templates. Peeling of the soft template from the matrix is shown in the inset. Scale bar, 100 μm . **f**, Comparison of the pull-out force of the rigid templates and the soft templates.

Fig. 4 Microchannel structures fabricated by soft demoulding. **a**, The thinnest channel (10 μm in diameter) fabricated in this work. Scale bar, 50 μm . **b** and **c**, The microchannel and its circular cross-section. Scale bars, 50 μm . **d**, The microchannel with tapered geometry. The diameter of the left side of the tapered channel is 250 μm , and the right-side diameter is 40 μm . Scale bar, 200 μm . **e**, The microchannel with spindle-knotted shape. Scale bar, 500 μm . **f**, The microchannel lattice. Scale bar, 500 μm . **g**, The helical microchannel. Scale bar, 200 μm . **h**, The branched microchannel. Scale bar, 100 μm . **i**, The microchannel with plectoneme structure. Scale bar, 500 μm . **(j)** and **(k)** are the feature parts of **(f)** and **(i)**, respectively. Scale bars, 200 μm . **l-q**, The novel 3D microchannel structures. Both **(n)** and **(o)** show the different views of a same prototype, and the same to **(p)** and **(q)**. Scale bars, 5 mm. **r**, Feature size and geometric flexibility for microchannel fabrication studies (Assembly capability is not considered for comparison).

Supplementary Fig. 4 Characterization of soft demoulding. **a**, The peel model for soft demoulding. **b**, The angle θ and the stretch force changing with strain for the soft template (thermoplastic resin). **c**, The effect of soft template (thermoplastic resin) diameter on the peel force and peel angle. **d**, The stress-strain curve for a soft template (TPU).

RC: (3) The claim of "snake shedding inspired" seems not firmly supported in the manuscript. It is a nice way of explaining the process. However, there is no theoretical or experimental support to justify the claim. There is no citation of references to support the claim either. The authors are suggested to back up the claim by elaboration and/or references or to remove the claim. The work is significant by its own right, without the need for a "nature-inspired" tag.

AR: The authors thank the reviewer and editor's suggestions, and we decided to remove the "nature-inspired" tag in this work.

We mainly revised the title and deleted the bioinspired description in the 1st paragraph of "Abstract" section, 3rd paragraph of "Introduction" section, and revised Fig. 1 as follows:

"Soft demoulding for high-complexity ultra-thin microchannels"

"Here, we propose a simple, solvent-free microchannel fabrication method capable of producing monolithic microchannels with complex 3D structures, long length, and small diameter."

"Inspired by the tension-induced necking phenomenon during the cold drawing process of the polymeric specimens^{32, 33}, we propose a simple, fast, solvent-free method to generate micro-level 3D monolithic channels (Supplementary Fig. 1)."

Fig. 1 The concept and mechanism of soft demoulding. **a**, The conceptual schematic of soft demoulding and its typical applications. **b**, The soft demoulding process: the template is first embedded in the matrix, then it is stretched and peeled out, and consequently, the channel is formed.

RC: (4) The description of the work (from page 12) also requires further clarifications.

(a) It is often not clear whether the authors used 3748Q or TPU-95A. Please specify the material used for each experiment.

AR: We clarified the materials information in all the relative sections as follows:

In the 3rd paragraph of "Methods" section:

"All the soft templates were made of hot melt adhesive 3748Q."

In the 4th paragraph of "Methods" section:

"The 10 μm microchannel (Fig. 4a) was fabricated by the TPU filament. The other microchannel structures shown in Fig. 4 and Supplementary Fig. 5 were fabricated by the hot melt adhesive 3748Q."

In the 8th paragraph of "Methods" section:

"The soft worm robot was fabricated using silicone (E610, Shenzhen Hong Ye Jie Technology Co.) with a thermoplastic resin template (hot melt adhesive 3748Q)."

In the 9th paragraph of "Methods" section:

"The soft tendril robot was created by first embedding a long helical TPU template in the silicone matrix (Ecoflex 0050, Smooth-on) and peeling the template out."

In the 10th paragraph of "Methods" section:

"First, a straight microchannel (length: 20 cm and diameter: 150 μm) was generated in the silicone matrix (Ecoflex 0050, Smooth-on) by a TPU filament."

In the 11th paragraph of "Methods" section:

"The soft helical template was generated by fixing a straight TPU filament onto a metal cone and heating the filament with a heat gun (200 °C) to reach the various diameter helical soft template."

In the 13th paragraph of "Methods" section:

"First, the soft filaments (Hot melt adhesive, 3748Q) were fixed to a designed 3D printed mould."

In the 15th paragraph of "Methods" section:

"After 6 hours of incubation, the soft template (Hot melt adhesive, 3748Q) was demoulded from the gel to generate the microchannel."

RC: (b) In page 13, comparisons were made to emphasize the advantage of the present work. First of all, this should be a part of the main text, not the Method. Second, this work is strongly influenced by the work by Verma et al. So, Comparison with the use of nylon templates seems informative, and even imperative.

AR: The authors thank the reviewer's valuable suggestions. We moved the comparison part to the 3rd and 4th paragraphs of "Results" section as follows:

"There is competition between the critical fracture force of the rigid template and the shear force during pulling. Once the shear force is larger than the critical fracture force, the template

fractures and the demoulding fails. Only when the critical fracture force is larger, the template can be pulled out. Assuming the template is a simple straight wire with a round cross-section, the shear force F_{shear} is determined by:

$$F_{shear} = \tau \pi d l \quad (2)$$

where τ is the shear stress, d is the filament diameter, and l is the embedded length. The shear force increases linearly with the diameter and the embedded length of the template, as shown in Fig. 3 b. The template's fracture force F_{frac} is calculated as:

$$F_{frac} = \sigma_f A = \frac{\sigma_f \pi d^2}{4} \quad (3)$$

where σ_f is the fracture stress, and A is the cross-sectional area of the template. Hence, a longer embedded length demands a thicker diameter for successful extraction. To avoid fracture, we have $F_{shear} \leq F_{frac}$, and thus:

$$\frac{l}{d} \leq \frac{\sigma}{4\tau} \quad (4)$$

which indicates that the aspect ratio of the channel is intrinsically limited by the nature of the rigid template materials, i.e., the strength and adhesive energy density. For instance, the maximum aspect ratio for the PDMS matrix and the copper template system is 258, according to equation (4) ($\sigma=6300$ MPa, $\tau= 6.1$ MPa (based on experimental results in this work)). As shown in Fig. 3c and Supplementary Fig. 3 b and c, when the embedded length was 30 mm, the copper (diameter: 80 μm) and nylon template (diameter: 100 μm) ruptured due to the large aspect ratios (375 and 300, respectively). Such a strong force might also break a fragile matrix (e.g., agarose gels). Moreover, a rigid template inevitably causes wear on the channel surfaces due to the rigidity of the template and the large shear force.

In contrast, for the soft demoulding method, peeling instead of shearing is dominant during the extraction process due to the larger stretching ratio of the template (the ultimate strain of the thermoplastic resin (1200%) was 100 times that of the copper wire (12%)), as shown in Fig. 3a). The aspect ratio of the channel no longer has a limit because the large deformation of the soft template transfers the demoulding mechanism to a peeling process (see "Methods" section 'Fabrication of the microchannels by soft demoulding'), and the peel force has no relationship with the embedded length of the soft template (Fig. 3 - d and f). The peel force F_{peel} for soft demoulding of thermoplastic resin (hot melt adhesive, 3748Q) can be expressed by⁴⁰:

$$F_{peel} = \Delta E_s \frac{\pi d}{(1 - \cos\theta)} \quad (5)$$

where d is the filament diameter, θ is the peel angle, and ΔE_s is the adhesive energy.

The peel force remained consistent for the samples with different embedded lengths (10 mm, 20 mm, 30 mm, and 40 mm) (Fig. 3e and f), indicating that the mechanism of soft demoulding was significantly different from the rigid demoulding shown previously (see "Methods" sections 'Mechanical characterization' and 'Demoulding tests'). As shown in Supplementary Fig. 4 a and b, when subjected to a stretching force, the strain stably increases, and the peel angle expands, while the force first surges under a short strain and then remains stable over a large range for the simulation results (see "Methods" section 'Simulation of the deformed angle'). It is observed in our soft demoulding experiments that the onset of peeling occurs when the peel force reaches a plateau (Supplementary Fig. 4c). For larger diameters, the peel force increases, and the consequent peel angles are different (Fig. 3d and Supplementary Fig. 4c). Demoulding failure always occurs in the situation where the template reaches its maximum strain before demoulding is initiated. According to the simulation, for the soft template (thermoplastic resin) in this work, demoulding fails when the diameter is less than 15.1 μm , as shown in Fig. 3d, since the peel angle exceeds its largest value (30 degrees)."

Yes, this paper shares the same motivation as the work by Verma et al.—to create monolithic structures of microchannels with complex geometries. To avoid the large extracting force on the template, the work by Verma et al. chooses a smart way by 1) reducing the adhesive force using low interfacial energy materials, i.e., nylon as the template and PDMS as the matrix; 2) actively deforming the PDMS matrix by swelling. However, this method limits the matrix materials to solvent absorbing materials, i.e., PDMS, and swelling is uncontrollable and typically results in unfavorable buckling. Comparably, the soft demoulding proposed in our work reduces the extracting force by deforming the template. Thus, various materials are available for the template only if they are deformable, and the matrix material is not limited to PDMS, either.

To verify the difference, we tested the performance of the nylon template as a comparison. Although the maximum aspect ratio of the nylon template can approach approximately 300 (30 mm embedded length and 100 μm diameter, as shown in Supplementary Fig. 4b and c) due to its high fracture strength (Supplementary Fig. 4a) during the rigid demoulding process, it is still much less than the maximum aspect ratio of soft demoulding (6000).

We explained the above discussions in the 2nd of "Introduction" section, 3rd paragraphs of "Results" section, and Supplementary Fig. 3 as follows:

"Matrix swollen methods require the swelling and deswelling process of matrices for the demoulding process, which will cause buckling of the matrices and solvent residual."

"As shown in Fig. 3c and Supplementary Fig. 3 b and c, when the embedded length was 30 mm, the copper (diameter: 80 μm) and nylon template (diameter: 100 μm) ruptured due to the large

aspect ratios (375 and 300, respectively). Such a strong force might also break a fragile matrix (e.g., agarose gels)."

Supplementary Fig. 3 Characterization of rigid demoulding of nylon filaments. a, The stress-strain curve for a nylon filament. **b**, The relationship between pull-out force and embedded length. **c**, The pull-out force-displacement curve of nylon templates with different embedded lengths.

RC: (c) In Line 396 (page 15), the natural question is "What will happen to thermoplastic urethane if it were subjected to this modeling?" Please address this issue in the revision.

AR: For TPU, elastic potential energy should be considered in the equation due to the hyper-elastic behaviour. Then we have the relationship between peel force F_{peel} and peel angle θ as follows:

$$\frac{2F_{peel}^2}{Ed} + \frac{F_{peel}(1 - \cos\theta)}{\pi d} = \Delta E_s$$

where d is the filament diameter, E is Young's modulus and ΔE_s is the adhesion energy.

We complemented the stress-strain test and the demoulding experiments for the TPU filament of different embedded lengths (10 mm, 20 mm, 30 mm, and 40 mm). The TPU filament exhibited a hyper-elastic behaviour during the stretch-strain test (see Supplementary Fig. 4d). As the embedded length increased, both the pull-out force and the applied strain increased accordingly, as shown in Supplementary Fig. 4 e and f. Compared to the thermoplastic resin, the increment of the pull-out force is caused by the hyper-elastic property of the soft filament and the increasing strain during the demoulding process, as shown in Supplementary Fig. 4f. Based on our simulation results, as the strain increased on the TPU filament, both the pull-out force and the deformed angle increased accordingly, as shown in Supplementary Fig. 4g. Similar to the thermoplastic resin, the intersection points of peel force and applied force in Supplementary Fig. 4h exhibited the initial peel angles for the TPU filaments with different diameters (0.4 mm, 0.6 mm, and 0.8 mm). The results showed that the peel angle increased with the diameter increasing.

The corresponding discussion was added in the Supplementary Note 1 and Supplementary Fig. 4 as follows:

"Supplementary Note 1: Soft demoulding mechanism for the TPU filament

We built the demoulding model for the TPU filament. Due to the hyper-elastic behaviour of the TPU filament (Supplementary Fig. 4d), the relationship between peel force F_{peel} and peel angle θ can be expressed by:

$$\frac{2F_{peel}^2}{Ed} + \frac{F_{peel}(1 - \cos\theta)}{\pi d} = \Delta E_s \quad (1')$$

where E is Young's modulus, and d is the filament diameter.

Different from thermoplastic resin (hot melt glue, 3748Q), the TPU filaments exhibited an increment in peel force with embedded length increasing since the higher strain was engaged during the demoulding processes (see Supplementary Fig. 4e and f). The deformed angle and applied force enlarge as the strain increases (Supplementary Fig. 4g). In addition, for larger diameters, the peel force increases, and the peel angles increase accordingly (Supplementary Fig. 4h).

Compared to rigid demoulding, an increment in peel force is also observed in the demoulding of the TPU filament with the embedded length increasing, but the applied force is much smaller. Different from the effect of the contact area on the shear force of rigid demoulding, the TPU filament demoulding still obeys the soft demoulding mechanism (see equation (1')), and the increasing peel force can be explained by the increasing strain, as shown in Supplementary Fig. 4f. Moreover, the TPU filament is reusable until it exceeds its elastic region.

Supplementary Fig. 4 Characterization of soft demoulding. **c**, The effect of soft template (thermoplastic resin) diameter on the peel force and peel angle. **d**, The stress-strain curve for a soft template (TPU). **e**, The pull-out force-displacement curve of TPU templates with different embedded lengths. **f**, The effect of embedded length of TPU filament on the pull-out force and strain. **g**, The angle θ and the stretch force changing with strain for the TPU filaments. **h**, The effect of the TPU filaments' diameter on the peel force and peel angle.

RC: (d) The "bulge" (Fig. 2f, for example) can invalidate the "shedding" process described in Fig. 1. So, the reviewer anticipates a certain size limit in the bulge which restricts the utilization of the proposed technique. Please comment on this possibility.

AR: Yes, there is a limitation on the bulge size – its diameter after deformation should be smaller than the channel. We added a new section in the supplementary material (Supplementary Note 2) and Supplementary Fig. 10 to discuss this aspect in detail as follows:

"Supplementary Note 2: Demoulding of the filament with a bulge

As shown in Fig. 2f, there is a limitation on the bulge size—after deformation, the bulge size should be smaller than other sections of the channel.

Assuming that the elasticity of the filament material is linear and the length of the bulge is much smaller than the filament length, for a uniform filament (Supplementary Fig. 10a), we have the Poisson's ratio:

$$\nu = -\frac{\varepsilon_{trans}}{\varepsilon_{axial}} = \frac{R - r/R}{l - L/L} = \frac{R - r}{R\varepsilon} \quad (2')$$

where R is the original radius, r is the stretched radius, and ε is axial strain.

For the bulge filament (Supplementary Fig. 10b), the force is constant in every part of the filament during the stretch process:

$$F_{bulge} = F_{filament} \quad (3')$$

Then we have:

$$\frac{\sigma_1}{\sigma_2} = \frac{R_2^2}{R_1^2} \quad (4')$$

Since the material elasticity is linear:

$$\frac{\varepsilon_1}{\varepsilon_2} = \frac{\sigma_1}{\sigma_2} \quad (5')$$

Combining equation (3') -(5'), we get:

$$r_2 = R_2 - \frac{R_2^3 \varepsilon \nu}{R_1^2} \quad (6')$$

where R_1 is the original filament diameter, r_1 is the stretched filament diameter, R_2 is the original bulge diameter, and r_2 is the stretched bulge diameter.

To ensure that the bulge can pass through the microchannel, we should have:

$$R_1 \geq R_2 - \frac{R_2^3 \epsilon_V}{R_1^2} \quad (7')$$

Therefore, the bulge size is limited by the original filament diameter, Poisson's ratio, and the ultimate strain of the filament."

Supplementary Fig. 10 Illustrations of stretching of soft templates. **a**, Stretching of a single soft filament. **b**, Stretching of a single soft filament with a bulge.

RC: Overall, the manuscript describes a new and potentially useful microfabrication technique, but its presentation and application need to be greatly improved for publication in Nature Communications.

Some miscellaneous points include:

- (a) The sentence in Line 73~74 is repeated and redundant.
- (b) In Line 93, "Except for" seems to mean "In addition to".
- (c) Figure 3u seems too busy. Just use the last names.
- (d) Supplementary Fig. 4c doesn't seem to play any role.

AR: The authors appreciate the reviewer's comments.

Accordingly, we revised the draft as follows:

(a) We deleted the sentence:

"Here, we refer to this method as "soft demoulding". By varying the dimensions and geometries of the template..."

(b) We replaced "Expect for" with "in addition to":

"In addition to the 1D template, more complex profiles can be fabricated by..."

(c) We deleted the authors' first names and published years as follows:

(d) We deleted Supplementary Fig. 4c.

Reviewer #2 :

Paper summary: In this paper, the authors report a method for creating microfluidic channels using soft demolding, inspired by snake shedding. The soft demolding allowed them to create fairly complex, circular channel structures, with aspect ratios well above conventional values (>6000). As a result, the authors fabricated 3D microfluidic channels with flower-like opening and closing functions, microfluidic channels with skin-like sensor systems, and microfluidic channels with blood vessel models. In my opinion, the results of this paper are of interest to the broader community of microfluidics researchers working in different areas and are likely to be of great interest to the readers of Nature Communications. Therefore, in my opinion, this paper is worthy of publication, but it would be desirable to make some revisions in the points listed below.

RC: (1) I like how the authors consider ideas about the microfluidic fabrication method by soft demolding. However, I miss a control experiment to show that the vascular model obtained in their experiment cannot be obtained with a channel produced by soft lithography. In other words, what they have shown in Fig. 4f is reproduced by soft lithography without their technique. I think this is important to support their claim that "using soft lithography simply yields rectangular 2D channels and does not allow for ultra-thin, 3D complex structures".

AR: The authors thank the reviewer's comment. Soft lithography can fabricate elastic layers with grooves by replicating the mould made by photolithography. This method has several limitations. First, soft lithography cannot fabricate the monolithic channels, and most of the channels made by soft lithography are composed of two parts bonded together. Moreover, due to the exposure process (the UV light passes through the photomask), the typical shape made by photolithography is a rectangle⁴⁴. It is very challenging to generate a round shape (Hongbin, Guangya et al. 2009, Fiddes, Raz et al. 2010). In comparison, the soft demoulding method proposed in this work can generate monolithic complex-shaped (including round shape) microchannels.

According to the reviewer's suggestions, we redesigned the experiments on artificial vessels. We fabricated a monolithic artificial vessel with a thin round channel (diameter: 150 μm) and a tapered artificial vessel. With the same material (agarose gels), we also demonstrated that more complex structures (a spindle-shaped one and a narrow-necked one) were made by the soft demoulding technique.

The corresponding revision can be found in the 11th paragraph of "Results" section, Fig. 5, and Supplementary Fig. 9 as follows:

"Finally, using the soft demoulding technique, we created microchannels in the agarose gels and artificial blood vessels with straight and tapered structures."

"Here, we first fabricated a microchannel with a spindle-shaped structure and a microchannel with a narrow neck in agarose gels (Supplementary Fig. 9 a, b, and "Methods" section

'Fabrication of biocompatible microchannel structures'), which can be used for the vascular disease models. Furthermore, we built a round cross-sectional artificial vessel (diameter: 150 μm) and a tapered one (the minimum diameter: 250 μm , and the maximum diameter: 500 μm) by seeding human umbilical vein endothelial cells (HUVECs) into the microchannel within the fibrin gel matrix (see Fig. 5e, Supplementary Fig. 9 c-f, and "Methods" section 'Fabrication of the artificial vascular model')."

Fig. 5 Demonstrative applications of soft demoulding. e, The artificial blood vessels in the fibrin gel with HUVECs seeded. The confocal image of the cross-sectional views of the image (z-projection of a 250 μm stack) of the tapered artificial vessel (the minimum diameter: 250 μm , the maximum diameter: 500 μm) and the straight artificial vessel (diameter: 150 μm) after one day of HUVECs seeding. The confocal images of the fibrin gel after 1-2 days of culture stained with live (green)/dead (red) assay. Scale bars, 200 μm .

Supplementary Fig. 9 Applications on biomedical devices based on soft demoulding methodology. **a**, The spindle-shaped microchannel in the agarose gel (left diameter: 100 μm , the maximum diameter of the spindle shape: 450 μm , and right diameter: 200 μm). **b**, The straight microchannel with a narrow neck (diameter: 300 μm , and the narrow diameter: 150 μm). **e**, The brightfield image of the straight artificial vessels (diameter: 150 μm) in the fibrin gels formed by soft demoulding. **f**, The brightfield images of the top view and cross-sectional view of the tapered artificial vessels (the minimum diameter: 250 μm , the maximum diameter: 500 μm). Scale bars, 200 μm .

References

- R1. Fiddes, L. K., et al. (2010). "A circular cross-section PDMS microfluidics system for replication of cardiovascular flow conditions." *Biomaterials* **31**(13): 3459-3464.
- R2. Hongbin, Y., et al. (2009). "Novel polydimethylsiloxane (PDMS) based microchannel fabrication method for lab-on-a-chip application." *Sensors and Actuators B: Chemical* **137**(2): 754-761.

RC: (2) Specific reference to the need for complex 3D and high aspect ratio microfluidic channels is needed. The sentence "Most of the current fabrication methods are inadequate for biological applications that strictly require nontoxic and biocompatible elements" is unclear as to the reason for the need for complex 3D and high aspect ratio microfluidic channels.

AR: The authors appreciate this valuable suggestion. Three references have been added to the revised version. Becker et al. exhibited the importance of the high aspect ratio channel by showing the channel endowing the soft actuators with greater entanglement on target objects⁹. Xu et al. designed different optical laces by introducing optical fibers forming 3D complex geometries into 3D-printed matrices to mimic the complex afferent sensory neural network¹⁵, showing the importance of 3D complex channels. Grigoryan et al. fabricated complex 3D multi-vascular networks with high aspect ratio to reappear the function of alveoli¹⁷ and exhibited the significance for both high aspect ratio and 3D complex channels. Moreover, we added the limitations of the existing methods to better explain the sentence, "Most of the current fabrication

methods are inadequate for biological applications that strictly require nontoxic and biocompatible elements".

Accordingly, we revised the 1st and 2nd paragraphs of "Introduction" section as follows:

"For example, high-aspect-ratio channels endowed soft actuators with great entanglement for grasping^{9, 14}, and complex 3D optical laces were able to mimic the afferent sensory neural network¹⁵. High-aspect-ratio microchannels with 3D geometries are also critical to improving particle sorting efficiency¹⁶ and reappeared alveoli's function¹⁷."

"Additive manufacturing can generate 3D microchannels in intricate topological geometries, but the feature size and surface roughness are limited by the fabrication processes^{17, 20}. Matrix swollen methods require the swelling and deswelling process of matrices for the demoulding process, which will cause buckling of the matrices and solvent residual²⁸. Complex and ultra-thin microchannels can be fabricated by template dissolution methods, but dissolution and draining out become challenging due to the capillary effect when the channels are just tens of micrometers^{18, 29}. Other methods, such as employing liquid template³⁰ and laser processing technology³¹, also suffer limitations for 3D geometries and smooth channel generation. In addition, 3D microchannels assembly is also challenging due to templates fixing and removal processes."

RC: (3) Need a table summarizing soft lithography, emerging methods, and soft demolding with respect to 3D complexity, aspect ratio, channel geometry, toxicity, etc.

AR: According to this comment, we summarized and compared the microchannel fabrication ability in soft matrices among soft lithography, physical extraction, template dissolution, 3D printing, and soft demoulding, according to cross-sectional geometry, feature size, space complexity, aspect ratio, requiring solvent or not, roughness, and fabrication duration, as shown in Supplementary Table 1.

Fabrication method	Soft lithography	Physical extraction (Matrix swollen and direct pulling)	Template dissolution	3D printing (Direct printing channels)	Soft demoulding
Cross-sectional geometry	Rectangular shape	Round shape	Round shape	Round shape	Round shape
Feature size	10 μm ¹⁹	10 μm ²³ (matrix swollen) 100 μm ⁷ (direct pulling)	10 μm ²⁷	200 μm ¹⁷ , 100 μm ²⁰	10 μm
Space complexity	High 2D complexity	Medium 3D complexity	High 3D complexity	High 3D complexity	High 3D complexity

Aspect ratio	N/A	629 ⁹	N/A	N/A	6000
Solvent-required	No	Yes	Yes	No	No
Channel surface roughness	High smooth	High smooth	High smooth	Relative rough	High smooth
Fabrication duration	Several hours	Several hours	Half-day-Several days	Several hours	Several hours

Supplementary Table 1. Comparisons of existing microchannel fabrication technologies.

RC: (4) Interesting results on the miniaturization and complexity of a single structure are presented, but the question remains on the complexity of the aggregate. The current state of the art is shown in microchannel fabrication in Fig. 3u, but it seems to be due to the lack of explanation on the difficulty of assembling these structures, although the shape at the level of a single structure is mentioned.

AR: In the last version, the state of the art shown in Fig. 4u (i.e., Fig. 4r in the revised version) is based on single structures to facilitate the comparison. Assembly of multiple simple structures into 3D geometries is undoubtedly challenging.

In the new version, we rephrased the description of the state of the art in the 2nd paragraph of "Introduction" section and the legend of Fig. 4r as follows:

"In addition, 3D microchannels assembly is also challenging due to templates fixing and removal processes."

"Feature size and geometric flexibility for microchannel fabrication studies (Assembly capability is not considered for comparison)."

RC: (5) The merits of this paper are (1) a high aspect ratio and (2) the ability to fabricate structures of small size, and the results of each were shown. There was no mention, however, of (1) and (2) being necessary at the same time, and we felt the need to mention this.

AR: Thanks for the reviewer's valuable suggestions. Devices with both small dimensions and a high aspect ratio microchannel are essential for different applications. For example, they have been utilized for particle sorting with a 3D helical structure in millimeter to centimeter scale in Xi's work¹⁶ and for the artificial alveoli model building in Grigoryan's work¹⁷.

We revised the 1st paragraph of "Introduction" section as follows:

"High-aspect-ratio microchannels with 3D geometries are also critical to improving particle sorting efficiency¹⁶ and reappeared alveoli's function¹⁷."

RC: (6) In the case of fabricating assembled structures in combination with existing techniques, I felt that the size of the structures is several hundred μm , which obscures the advantages of their results. In addition, it is necessary to follow up on the mechanical resistance of the mold in order to expand the advantages of this method to "high aspect ratio" and/or "ultra-thin" microchannels.

AR: The authors thank the reviewer's two suggestions. The first suggestion is that the size of the channel structure (several hundred microns in the last version) should be smaller. In our new version, we fabricated new prototypes with thinner and complex microchannels, including a soft tendril robot containing a helical microchannel (diameter: 150 μm), a soft antenna containing a 3D helical microchannel (diameter: 180 μm) with variable diameter, a spindle-shaped microchannel (minimum diameter: 100 μm), a narrow-necked microchannel (minimum diameter: 150 μm) in agarose gels, and a thin artificial vessel (diameter: 150 μm).

Accordingly, we added the above information in the main text, Fig. 5, Supplementary Fig. 6, 8, and 9.

We revised the 1st paragraph of "Abstract" section as follows:

"We demonstrate the vast applicability and significant impact of this technology in multiple scenarios, such as a tendril robot capable of winding by a super-long helical microchannel, a soft mechanically-tunable miniature antenna with a 3D helical microchannel, and artificial vessels with straight and tapered structures capable of transporting nutrition to the surrounding cells."

We also revised the description in the 8th paragraph of "Results" section:

"We also demonstrated a tendril-inspired ultra-long, soft robot (Fig. 5b, Supplementary Fig. 6b, and Supplementary Movie 3) integrating a helical microchannel (diameter: 150 μm) with an aspect ratio of more than 1600 (see "Methods" section 'Fabrication of the soft tendril robot'). Pressured by air, the soft tendril robot winded around a rod accordingly, which can be applied for fixation and grasping, such as fixing and monitoring nerval activity⁵¹, by imitating the tendril's survival strategy."

In the 10th paragraph of "Results" section:

"Moreover, a soft and mechanically tunable micro-antenna with a 3D helical conductive structure was fabricated by soft demoulding (see "Methods" section 'Fabrication of the soft antenna'). Existing small antennas, although critical for wearable devices, human-machine communication systems, and implant devices⁵³, are constricted by simple structures, such as

rodlike or planar geometries^{54, 55}, or supports for the 3D structures^{55, 56}. In addition, the large size (several centimeters) and rigid frame of current antennas hinder them from more extensive applications. Here, we fabricated a soft micro-antenna (10 mm x 1.2 mm x 1.2 mm) containing a 3D helical microchannel (channel diameter: 180 μm , helical structure diameter: 450 μm at the minimum and 900 μm at the maximum) infused with the liquid metal (Fig. 5d and Supplementary Fig. 8), while most previous small antennas were in the centimeter scale^{53, 56}. The 3D conducting structure offers a more compact dimension for confined environments. Moreover, with the 3D helical structure, the antenna presents low reflection coefficients of -6.6 and -22.3 dB (lower than -10 dB is sufficient for commercial antennas⁵⁷) at two resonant frequencies (6.8 GHz and 13.1 GHz), respectively. Being bent, the micro-antenna is mechanically tunable in two broad resonant frequency ranges (from 6.8 GHz to 7.3 GHz and 11.9 to 13.1 GHz), and the reflection coefficient becomes lower for better signal transmission, as shown in Fig. 5d (see "Methods" section 'Reflection coefficient test for the soft antenna'). Therefore, the fabrication method of soft demoulding for micro-antenna provides a new approach toward compact soft wireless electronics."

In the 11th paragraph of "Results" section:

"Here, we first fabricated a microchannel with a spindle-shaped structure and a microchannel with a narrow neck in agarose gels (Supplementary Fig. 9 a, b, and "Methods" section 'Fabrication of biocompatible microchannel structures'), which can be used for the vascular disease models. Furthermore, we built a round cross-sectional artificial vessel (diameter: 150 μm) and a tapered one (the minimum diameter: 250 μm , and the maximum diameter: 500 μm) by seeding human umbilical vein endothelial cells (HUVECs) into the microchannel within the fibrin gel matrix (see Fig. 5e, Supplementary Fig. 9 c-f, and "Methods" section 'Fabrication of the artificial vascular model')."

In the 12th and 13th paragraphs of "Results" section:

"We demonstrate its extensive applications by multiple prototypes, including a soft worm robot with a plectoneme-shaped structure, an ultra-long tendrill robot containing a helical microchannel (diameter: 150 μm , aspect ratio: more than 1600), a thread-like biocompatible wearable sensor, a soft antenna containing 3D helical microchannel with variable diameter, and a thin artificial blood vessel in tapered geometry. Our soft demoulding method also offers a promising approach to the microfluidics field and tissue engineering.

Although our soft demoulding method shows advantages for fabricating 3D complex microchannels with a high aspect ratio, the materials and fabrication method for soft templates employed in this work may limit it for more intricate and delicate microchannels generation. Future research will focus on the improvement of the design (e.g., computer-assisted design⁵⁹), fabrication process (e.g., employing high-precision moving stages for direct ink writing³⁴, and electrospinning technology⁶⁰), and materials (e.g., hydrogels, which show ultra-large stretch, toughness, and self-lubrication properties⁶¹) for soft template generation and matrix formation to

enhance the precision, complexity, versatility, and biocompatibility of the methodology for more extensive applications."

In the 11th, 13th, and 15th paragraphs of "Methods" section:

"Fabrication of the soft antenna. The soft helical template was generated by fixing a straight TPU filament onto a metal cone and heating the filament with a heat gun (200 °C) to reach the various diameter helical soft template. Then, this soft helical template was first fixed to a mould, and the elastomer precursor (Ecoflex 0050, Smooth-on) was poured into the mould. After the precursor was cured, the microchannel was generated by pulling the template out. Next, a syringe injected the liquid metal (-19°C, Dingguan Metal Technology Co.) into the microchannel. Finally, both ends of the microchannel were sealed by the silicone precursor."

"Fabrication of biocompatible microchannel structures. With soft demoulding, which is solvent-free and gentle in force, we fabricated different biocompatible microchannels from an aqueous agarose solution (1.5% w/v). First, the soft filaments (Hot melt adhesive, 3748Q) were fixed to a designed 3D printed mould. Then, the liquid agarose was poured into the mould to immerse the templates. After the agarose solidified, the agarose matrix was separated from the mould. Finally, the soft templates were demoulded from the agarose gels, and a microchannel lattice was formed, as shown in Supplementary Fig. 9 a and b."

"The mould for the artificial vessel was first immersed into 75% ethanol overnight and sterilized with ultraviolet (UV) light for one hour before use. Fibrin gel (21.5 mg/mL) was formed by dissolving fibrinogen in DMEM with high glucose (4.5 g/L, Gibco), 10% FBS, and 1% penicillin-streptomycin. The centrifugation (1500-2000g) was applied to remove air bubbles. The 3T3 cell pellet was gently resuspended in the fibrinogen solution to a concentration of $5-10 \times 10^5$ cells/mL. As the cells were uniformly mixed with fibrinogen, thrombin was added to a final concentration of 3 U/mL. The HUVEC suspension was concentrated to 1.5×10^6 cells/mL and seeded into the channels. The fibrin gel mixture was quickly added to the model and placed in an incubator for crosslinking. After 6 hours of incubation, the soft template (Hot melt adhesive, 3748Q) was demoulded from the gel to generate the microchannel. HUVECs suspension was concentrated to 6×10^6 cells/mL, and first seeded into the bottom of the channel. After one hour of static culturing, the device was turned over, and the HUVECs suspension was seeded into the top of the channel with another 2 hours of incubation (Supplementary Fig. 9c). The non-adherent cells and other cell debris within the microchannel were removed by fresh medium. The mould was cultured with ECM for 8 hours under static conditions to allow the cells to adhere and spread before introducing hemodynamic flow by rocking platform (Supplementary Fig. 9d)."

Fig. 5 Demonstrative applications of soft demoulding. **a**, The pneumatic soft robot in a twisting state mimics the millipede in a defence state (left inset) using the plectoneme microchannel. Scale bars (a-d), 5 mm. **b**, The long, soft tendrill robot (length: 10 cm) containing a helical microchannel (diameter: 150 μm) (aspect ratio > 1600) climbs on a rod after being inflated, like the real tendrill (left inset). **c**, The soft, thin, long strain sensor (channel diameter: 150 μm , length: 15 cm) capable of acquiring the elbow motion. **d**, The soft antenna containing a 3D helical microchannel (diameter: 180 μm) exhibiting different reflection coefficients under different deflection. Scale bar (inset): 200 μm . **e**, The artificial blood vessels in the fibrin gel with HUVECs seeded. The confocal image of the cross-sectional views of the image (z-projection of a 250 μm stack) of the tapered artificial vessel (the minimum diameter: 250 μm , the maximum diameter: 500 μm) and the straight artificial vessel (diameter: 150 μm) after one day of HUVECs seeding. The confocal images of the fibrin gel after 1-2 days of culture stained with live (green)/dead (red) assay. Scale bars, 200 μm .

Supplementary Fig. 6 Applications of soft demoulding on soft robotics for shape morphing. **a**, The structure of the soft worm robot. Scale bar, 500 μm . **b**, The structure of the soft tendril robot and its winding state after inflation. The left inset shows the structure of the helical microchannel (scale bar: 200 μm). Scale bar, 5 mm.

Supplementary Fig. 8 Applications of the soft antenna. **a**, Fabrication process of the soft antenna with a 3D helical microchannel. A 3D helical soft filament is fabricated by first aligning a straight soft filament to a metal cone into 3D geometry and fixing the helical geometry by heat treatment. Then, the soft template is immersed into the liquid precursor and thermally cured. Next, the templates are demoulded from the elastic matrix. Finally, the liquid metal is injected into the helical microchannel. **b**, The setup for the reflection coefficient tests.

Supplementary Fig. 9 Applications on biomedical devices based on soft demoulding methodology. **a**, The spindle-shaped microchannel in the agarose gel (left diameter: 100 μm , the maximum diameter of the spindle shape: 450 μm , and right diameter: 200 μm). **b**, The straight microchannel with a narrow neck (diameter: 300 μm , and the narrow diameter: 150 μm). **e**, The brightfield image of the straight artificial vessels (diameter: 150 μm) in the fibrin gels formed by soft demoulding. **f**, The brightfield images of the top view and cross-sectional view of the tapered artificial vessels (the minimum diameter: 250 μm , the maximum diameter: 500 μm). Scale bars, 200 μm .

The reviewer's second suggestion is to emphasize the mechanical resistance of the template during the demoulding process. We added more explanation on the soft demoulding modeling and discussion in the "Soft demoulding" part. Moreover, we conducted experiments on nylon filament (rigid template for rigid demoulding) for comparison and attached the discussion in the main text.

We added the above information in the 3rd and 4th paragraphs of "Results" section, Supplementary Note 1, Fig. 5, Supplementary Fig. 3, and Supplementary Fig. 4b as follows:

"There is competition between the critical fracture force of the rigid template and the shear force during pulling. Once the shear force is larger than the critical fracture force, the template fractures and the demoulding fails. Only when the critical fracture force is larger, the template can be pulled out. Assuming the template is a simple straight wire with a round cross-section, the shear force F_{shear} is determined by:

$$F_{shear} = \tau \pi d l \quad (2)$$

where τ is the shear stress, d is the filament diameter, and l is the embedded length. The shear force increases linearly with the diameter and the embedded length of the template, as shown in Fig. 3 b."

"As shown in Fig. 3c and Supplementary Fig. 3 b and c, when the embedded length was 30 mm, the copper (diameter: 80 μm) and nylon template (diameter: 100 μm) ruptured due to the large aspect ratios (375 and 300, respectively). Such a strong force might also break a fragile matrix (e.g., agarose gels). Moreover, a rigid template inevitably causes wear on the channel surfaces due to the rigidity of the template and the large shear force."

"Supplementary Note 1: Soft demoulding mechanism for the TPU filament

We built the demoulding model for the TPU filament. Due to the hyper-elastic behaviour of the TPU filament (Supplementary Fig. 4d), the relationship between peel force F_{peel} and peel angle θ can be expressed by:

$$\frac{2F_{peel}^2}{Ed} + \frac{F_{peel}(1 - \cos\theta)}{\pi d} = \Delta E_s \quad (1')$$

where E is Young's modulus, d is the filament diameter.

Different from thermoplastic resin (hot melt glue, 3748Q), the TPU filaments exhibited an increment in peel force with embedded length increasing since the higher strain was engaged during the demoulding processes (see Supplementary Fig. 4e and f). The deformed angle and applied force enlarge as the strain increases (Supplementary Fig. 4g). In addition, for larger diameters, the peel force increases, and the peel angles increase accordingly (Supplementary Fig. 4h).

Compared to rigid demoulding, an increment in peel force is also observed in the demoulding of the TPU filament with the embedded length increasing, but the applied force is much smaller. Different from the effect of the contact area on the shear force of rigid demoulding, the TPU filament demoulding still obeys the soft demoulding mechanism (see equation (1')), and the increasing peel force can be explained by the increasing strain, as shown in Supplementary Fig. 4f. Moreover, the TPU filament is reusable until it exceeds its elastic region.

Supplementary Fig. 3 Characterization of rigid demoulding of nylon filaments. **a**, The stress-strain curve for a nylon filament. **b**, The relationship between pull-out force and embedded length. **c**, The pull-out force-displacement curve of nylon templates with different embedded lengths.

Supplementary Fig. 4 Characterization of soft demoulding. **e**, The pull-out force-displacement curve of TPU templates with different embedded lengths. **f**, The effect of embedded length of TPU filament on the pull-out force and strain. **g**, The angle θ and the stretch force changing with strain for the TPU filaments. **h**, The effect of the TPU filaments' diameter on the peel force and peel angle.

Reviewer #3:

Paper summary: This paper by Fan et al. reports a fabrication method of microchannels. The authors say this method was inspired by snake-shedding, but I don't think this method is superior to other methods. Although it may be unique as a fabrication method, the structure fabricated in the authors' demonstration is not very useful. Fabricating simple microchannel (such as straight) is easy, but fabricating complex channel structures can be tedious to assemble soft templates. Moreover, accurate placement is not easy. I think any of the structures they have shown can be made without using this method. Therefore, I concluded that this paper has not reached the scientific level worthy of being published in Nature Communications.

AR: The authors thank the reviewer's comments. The reviewer has five questions in the comments, and here we reply to them one by one as follows:

1) I don't think this method is superior to other methods.

We did not mean this method is superior to other methods in all aspects (every method has its pros and cons). Still, this method is good at generating monolithic microchannels with 3D complex structures and high aspect ratios, which are challenging with previous methods but valuable in various scenarios, including soft actuators, sensory neural networks, particle sorting, and artificial alveoli. For example, in our demonstration, we fabricated a soft tendril robot containing a helical microchannel (aspect ratio over 1600) and a soft micro-antenna with a 3D helical microchannel (diameter: 180 μm) with variable diameter, spindle-shaped, and narrow-necked microchannels in agarose gels.

In comparison, the soft lithography technique suffers from limited cross-sectional shapes (rectangular) and spatial structures (two-dimensional (2D) patterns only), intensive labour, and expensive fabrication devices. It is also difficult to fabricate the 3D complex monolithic microchannels by the emerging technologies, e.g., additive manufacturing, matrix swollen, and template dissolution methods. For example, additive manufacturing can generate 3D microchannels in intricate topological geometries, but the feature size and surface roughness are limited. Matrix swollen methods require the swelling and deswelling process of matrices for the demoulding process, which causes buckling of the matrices and solvent residual. Complex and ultra-thin microchannels can be fabricated by template dissolution methods, but dissolution and draining out become challenging due to the capillary effect when the channels are just tens of micrometers. Compared to other technology, our soft demoulding is a simple, fast, solvent-free method for smooth and high-aspect-ratio monolithic microchannels with 3D complex structures.

We revised corresponding parts in the 2nd paragraph of "Introduction" section and added a comparison table to clarify the differences in Supplementary Table 1 as follows:

"The widely accepted soft lithography technique suffers from limited cross-sectional shapes (rectangular) and spatial structures (two-dimensional (2D) patterns only), intensive labour, and expensive fabrication devices, and it is unable to generate monolithic structures,^{6, 19.}"

"Additive manufacturing can generate 3D microchannels in intricate topological geometries, but the feature size and surface roughness are limited by the fabrication processes^{17, 20.} Matrix swollen methods require the swelling and deswelling process of matrices for the demoulding process, which will cause buckling of the matrices and solvent residual^{28.} Complex and ultra-thin microchannels can be fabricated by template dissolution methods, but dissolution and draining out become challenging due to the capillary effect when the channels are just tens of micrometers^{18, 29.} Other methods, such as employing liquid template³⁰ and laser processing technology^{31,} also suffer limitations for 3D geometries and smooth channel generation. In addition, 3D microchannels assembly is also challenging due to templates fixing and removal processes."

Fabrication method	Soft lithography	Physical extraction (Matrix swollen and direct pulling)	Template dissolution	3D printing (Direct printing channels)	Soft demoulding
Cross-sectional geometry	Rectangular shape	Round shape	Round shape	Round shape	Round shape
Feature size	10 μm^{19}	10 μm^{23} (matrix swollen) 100 μm^7 (direct pulling)	10 μm^{27}	200 μm^{17} , 100 μm^{20}	10 μm
Space complexity	High 2D complexity	Medium 3D complexity	High 3D complexity	High 3D complexity	High 3D complexity
Aspect ratio	N/A	629 ⁹	N/A	N/A	6000
Solvent-required	No	Yes	Yes	No	No
Channel surface roughness	High smooth	High smooth	High smooth	Relative rough	High smooth
Fabrication duration	Several hours	Several hours	Half-day- Several days	Several hours	Several hours

Supplementary Table 1. Comparisons of existing microchannel fabrication technologies.

2) Although it may be unique as a fabrication method, the structure fabricated in the authors' demonstration is not very useful.

The authors appreciate the reviewer's comments. It is true that our prototypes in the demonstration cannot be employed as products immediately or commercialized immediately, but this work focuses more on the fundamental fabrication mechanism for broad applications. To show the potentially extensive applications in the manuscript, we fabricated various structures for the actuators, sensors, and biological devices in the last version.

To further strengthen the comprehensive employment of the technique, in our new version, we redesigned and fabricated a soft tendril robot containing a 3D high-aspect-ratio helical microchannel (diameter: 150 μm , length: 25 cm) with a cross-sectional area of 1 mm^2 and achieved winding behavior just like real tendrils. This miniature robot as a platform can benefit various surgical operations, such as nerve fixing and monitoring⁵¹. We also fabricated a mechanically tunable soft micro-antenna with the 3D helical microchannel (channel diameter: 180 μm , helical structure diameter: 450 μm at the minimum and 900 μm at the maximum), which shows great potential applications in flexible electronics⁵³.

Moreover, our soft demoulding is applicable for artificial vessels due to the solvent-free and delicate fabrication process. We supplemented new experiments on the microchannels with spindle-shaped and narrow-necked structures in agarose gels and the artificial vascular models with a thin straight artificial vessel (diameter: 150 μm) and a tapered artificial vessel. This vascular structure can be used for vessel growth and further vascular disease prediction⁴.

In detail, we added the above information in the main text, Fig. 5, Supplementary Fig. 6, 8, and 9.

We revised the 1st paragraph of "Abstract" section as follows:

"We demonstrate the vast applicability and significant impact of this technology in multiple scenarios, such as a tendril robot capable of winding by a super-long helical microchannel, a soft mechanically-tunable miniature antenna with a 3D helical microchannel, and artificial vessels with straight and tapered structures capable of transporting nutrition to the surrounding cells."

We also revised the description in the 8th paragraph of "Results" section:

"We also demonstrated a tendril-inspired ultra-long, soft robot (Fig. 5b, Supplementary Fig. 6b, and Supplementary Movie 3) integrating a helical microchannel (diameter: 150 μm) with an aspect ratio of more than 1600 (see "Methods" section 'Fabrication of the soft tendril robot'). Pressured by air, the soft tendril robot winded around a rod accordingly, which can be applied for fixation and grasping, such as fixing and monitoring nerval activity⁵¹, by imitating the tendril's survival strategy."

In the 10th paragraph of "Results" section:

"Moreover, a soft and mechanically tunable micro-antenna with a 3D helical conductive structure was fabricated by soft demoulding (see "Methods" section 'Fabrication of the soft antenna'). Existing small antennas, although critical for wearable devices, human-machine communication systems, and implant devices⁵³, are constricted by simple structures, such as rodlike or planar geometries^{54, 55}, or supports for the 3D structures^{55, 56}. In addition, the large size (several centimeters) and rigid frame of current antennas hinder them from more extensive applications. Here, we fabricated a soft micro-antenna (10 mm x 1.2 mm x 1.2 mm) containing a 3D helical microchannel (channel diameter: 180 μm , helical structure diameter: 450 μm at the minimum and 900 μm at the maximum) infused with the liquid metal (Fig. 5d and Supplementary Fig. 8), while most previous small antennas were in the centimeter scale^{53, 56}. The 3D conducting structure offers a more compact dimension for confined environments. Moreover, with the 3D helical structure, the antenna presents low reflection coefficients of -6.6 and -22.3 dB (lower than -10 dB is sufficient for commercial antennas⁵⁷) at two resonant frequencies (6.8 GHz and 13.1 GHz), respectively. Being bent, the micro-antenna is mechanically tunable in two broad resonant frequency ranges (from 6.8 GHz to 7.3 GHz and 11.9 to 13.1 GHz), and the reflection coefficient becomes lower for better signal transmission, as shown in Fig. 5d (see "Methods" section 'Reflection coefficient test for the soft antenna'). Therefore, the fabrication method of soft demoulding for micro-antenna provides a new approach toward compact soft wireless electronics."

In the 11th paragraph of "Results" section:

"Here, we first fabricated a microchannel with a spindle-shaped structure and a microchannel with a narrow neck in agarose gels (Supplementary Fig. 9 a, b, and "Methods" section 'Fabrication of biocompatible microchannel structures'), which can be used for the vascular disease models. Furthermore, we built a round cross-sectional artificial vessel (diameter: 150 μm) and a tapered one (the minimum diameter: 250 μm , and the maximum diameter: 500 μm) by seeding human umbilical vein endothelial cells (HUVECs) into the microchannel within the fibrin gel matrix (see Fig. 5e, Supplementary Fig. 9 c-f, and "Methods" section 'Fabrication of the artificial vascular model')."

In the 12th and 13th paragraphs of "Results" section:

"We demonstrate its extensive applications by multiple prototypes, including a soft worm robot with a plectoneme-shaped structure, an ultra-long tendril robot containing a helical microchannel (diameter: 150 μm , aspect ratio: more than 1600), a thread-like biocompatible wearable sensor, a soft antenna containing 3D helical microchannel with variable diameter, and a thin artificial blood vessel in tapered geometry. Our soft demoulding method also offers a promising approach to the microfluidics field and tissue engineering."

Although our soft demoulding method shows advantages for fabricating 3D complex microchannels with a high aspect ratio, the materials and fabrication method for soft templates employed in this work may limit it for more intricate and delicate microchannels generation. Future research will focus on the improvement of the design (e.g., computer-assisted design⁵⁹), fabrication process (e.g., employing high-precision moving stages for direct ink writing³⁴, and electrospinning technology⁶⁰), and materials (e.g., hydrogels, which show ultra-large stretch, toughness, and self-lubrication properties⁶¹) for soft template generation and matrix formation to enhance the precision, complexity, versatility, and biocompatibility of the methodology for more extensive applications."

In the 11th, 13th, and 15th paragraphs of "Methods" section:

***"Fabrication of the soft antenna.** The soft helical template was generated by fixing a straight TPU filament onto a metal cone and heating the filament with a heat gun (200 °C) to reach the various diameter helical soft template. Then, this soft helical template was first fixed to a mould, and the elastomer precursor (Ecoflex 0050, Smooth-on) was poured into the mould. After the precursor was cured, the microchannel was generated by pulling the template out. Next, a syringe injected the liquid metal (-19°C, Dingshan Metal Technology Co.) into the microchannel. Finally, both ends of the microchannel were sealed by the silicone precursor."*

***"Fabrication of biocompatible microchannel structures.** With soft demoulding, which is solvent-free and gentle in force, we fabricated different biocompatible microchannels from an aqueous agarose solution (1.5% w/v). First, the soft filaments (Hot melt adhesive, 3748Q) were fixed to a designed 3D printed mould. Then, the liquid agarose was poured into the mould to immerse the templates. After the agarose solidified, the agarose matrix was separated from the mould. Finally, the soft templates were demoulded from the agarose gels, and a microchannel lattice was formed, as shown in Supplementary Fig. 9 a and b."*

"The mould for the artificial vessel was first immersed into 75% ethanol overnight and sterilized with ultraviolet (UV) light for one hour before use. Fibrin gel (21.5 mg/mL) was formed by dissolving fibrinogen in DMEM with high glucose (4.5 g/L, Gibco), 10% FBS, and 1% penicillin-streptomycin. The centrifugation (1500-2000g) was applied to remove air bubbles. The 3T3 cell pellet was gently resuspended in the fibrinogen solution to a concentration of $5-10 \times 10^5$ cells/mL. As the cells were uniformly mixed with fibrinogen, thrombin was added to a final concentration of 3 U/mL. The HUVEC suspension was concentrated to 1.5×10^6 cells/mL and seeded into the channels. The fibrin gel mixture was quickly added to the model and placed in an incubator for crosslinking. After 6 hours of incubation, the soft template (Hot melt adhesive, 3748Q) was demoulded from the gel to generate the microchannel. HUVECs suspension was concentrated to 6×10^6 cells/mL, and first seeded into the bottom of the channel. After one hour of static culturing, the device was turned over, and the HUVECs suspension was seeded into the top of the channel with another 2 hours of incubation (Supplementary Fig. 9c). The non-adherent cells and other cell debris within the microchannel were removed by fresh medium. The mould was cultured

with ECM for 8 hours under static conditions to allow the cells to adhere and spread before introducing hemodynamic flow by rocking platform (Supplementary Fig. 9d)."

Fig. 5 Demonstrative applications of soft demoulding. **a**, The pneumatic soft robot in a twisting state mimics the millipede in a defence state (left inset) using the plectoneme microchannel. Scale bars (a-d), 5 mm. **b**, The long, soft tendril robot (length: 10 cm) containing a helical microchannel (diameter: 150 μm) (aspect ratio >1600) climbs on a rod after being inflated, like the real tendril (left inset). **c**, The soft, thin, long strain sensor (channel diameter: 150 μm, length: 15 cm) capable of acquiring the elbow motion. **d**, The soft antenna containing a 3D helical microchannel (diameter: 180 μm) exhibiting different reflection coefficients under different deflection. Scale bar (inset): 200 μm. **e**, The artificial blood vessels in the fibrin gel with HUVECs seeded. The confocal image of the cross-sectional views of the image (z-projection of a 250 μm stack) of the tapered artificial vessel (the minimum diameter: 250 μm, the maximum diameter: 500 μm) and the straight artificial vessel (diameter: 150 μm) after one day of HUVECs seeding. The confocal images of the fibrin gel after 1-2 days of culture stained with live (green)/dead (red) assay. Scale bars, 200 μm.

Supplementary Fig. 6 Applications of soft demoulding on soft robotics for shape morphing. **a**, The structure of the soft worm robot. Scale bar, 500 μm . **b**, The structure of the soft tendril robot and its winding state after inflation. The left inset shows the structure of the helical microchannel (scale bar: 200 μm). Scale bar, 5 mm.

Supplementary Fig. 8 Applications of the soft antenna. **a**, Fabrication process of the soft antenna with a 3D helical microchannel. A 3D helical soft filament is fabricated by first aligning a straight soft filament to a metal cone into 3D geometry and fixing the helical geometry by heat treatment. Then, the soft template is immersed into the liquid precursor and thermally cured. Next, the template is demoulded from the elastic matrix. Finally, the liquid metal is injected into the helical microchannel. **b**, The setup for the reflection coefficient tests.

Supplementary Fig. 9 Applications on biomedical devices based on soft demoulding methodology. **a**, The spindle-shaped microchannel in the agarose gel (left diameter: 100 μm , the maximum diameter of the spindle shape: 450 μm , and right diameter: 200 μm). **b**, The straight microchannel with a narrow neck (diameter: 300 μm , and the narrow diameter: 150 μm). **e**, The brightfield image of the straight artificial vessels (diameter: 150 μm) in the fibrin gels formed by soft demoulding. **f**, The brightfield images of the top view and cross-sectional view of the tapered artificial vessels (the minimum diameter: 250 μm , the maximum diameter: 500 μm). Scale bars, 200 μm .

3) Fabricating simple microchannels (such as straight) is easy, but fabricating complex channel structures can be tedious to assemble soft templates.

The authors appreciate the reviewer's comments. It is true that fabricating short microchannels with a large diameter can be achieved by various other technologies, such as 3D printing or dip coating, but generating ultra-thin microchannels with a high aspect ratio and complex geometries is quite difficult. Of course, simple assembly of straight channels can generate various channels, but it has various limitations for complex structures, such as helical, spindle-knotted, and tree-like geometries.

Many researchers from different fields are exploring promising solutions to this multidisciplinary challenge. For example, just six months ago (November 2021, right after our submission of this paper), Nature published a new fabrication technology, bubble casting, to create "long, tortuous or vascular structures"¹⁴. In 2020, Becker et al.⁹ fabricated a super-high aspect ratio (629) channel for a slender, soft actuator, utilizing dip coating combining electrical fields or surface tension, but the channel diameter is 1

mm. In 2020, Kinstlinger et al.¹³ proposed a carbohydrate templates fabrication method for dendritic vascular network generating, while the surface roughness and diameter of templates are relatively large due to the laser sintering fabrication process. Therefore, the technologies for generating complex ultrathin microchannels are still burgeoning and challenging.

We revised the 1st and 2nd paragraphs of "Introduction" section as follows:

"For example, high-aspect-ratio channels endowed soft actuators with great entanglement for grasping^{9, 14}, and complex 3D optical laces were able to mimic the afferent sensory neural network¹⁵."

"Additive manufacturing can generate 3D microchannels in intricate topological geometries, but the feature size and surface roughness are limited by the fabrication processes^{17, 20}."

4) Moreover, accurate placement is not easy.

It is true that accurate placement for template assembly is not easy in this work since the template assembly is achieved by the constraint of moulds. In this work, we focused on elaborating a new microchannel fabrication technology (soft demoulding) and its mechanism for microchannels of high aspect ratios and complex structures. The template assembly can be improved by high-precision moving stages in the future. Moreover, we can utilize additive manufacturing for complex 3D soft template generation, which could ignore the template assembly process, to fabricate intricate microchannels with high precision.

To clarify the above information, we added sentences in the 2nd paragraph of "Introduction" section and the 2nd paragraph of "Discussion" section, respectively, as follows:

"In addition, 3D microchannels assembly is also challenging due to templates fixing and removal processes."

"Future research will focus on the improvement of the design (e.g., computer-assisted design⁵⁹), fabrication process (e.g., employing high-precision moving stages for direct ink writing³⁴, and electrospinning technology⁶⁰)."

5) I think any of the structures they have shown can be made without using this method.

The authors appreciate the reviewer's comment. We understand that the fabrication of microchannels is critical for various applications and has been widely studied for a long time. Still, previous methods fail when new needs have emerged recently with the development of technologies, such as soft devices.

As we know, currently, to fabricate microchannels, the primary methods are (1) soft lithography, (2) physical extraction, (3) template dissolution, and (4) 3D printing. For example, suppose we want a helical monolithic microchannel with a large aspect ratio (e.g., 1600, as shown in this work) in a soft matrix (e.g., PDMS, the most extensively applied materials in soft devices) for a soft micro-actuator, a soft sensor, a tunable antenna, or an artificial blood vessel. In that case, these methods fail due to different reasons. (1) Soft lithography, only available for open rectangle channels, can hardly generate the round monolithic channel in a 3D complex structure. (2) It is difficult for physical extraction (the maximum aspect ratio is 629 for a straight channel in the previous literature⁶) to remove the template out of the matrix with such a large aspect ratio since the high extracting force might break the template, particularly when the structure is a 3D complex geometry. (3) In template dissolution, the template is usually made by 3D printing, and the size is usually challenging to be tens of micrometers. Moreover, with such a large aspect ratio and thin diameter, the dissolution of the template and the draining out of the dissolved template are time-consuming and challenging with the concern of the capillary effect. (4) 3D printing is undoubtedly available in most scenarios, but in this case, it suffers from limited scalability for soft matter printing since high precision and large dimension are typically not available simultaneously.

We revised the corresponding parts in the 2nd paragraph of "Introduction" section as follows:

"The widely accepted soft lithography technique suffers from limited cross-sectional shapes (rectangular) and spatial structures (two-dimensional (2D) patterns only), intensive labour, and expensive fabrication devices, and it is unable to generate monolithic structures,^{6, 19}."

"Additive manufacturing can generate 3D microchannels in intricate topological geometries, but the feature size and surface roughness are limited by the fabrication processes^{17, 20}. Matrix swollen methods require the swelling and deswelling process of matrices for the demoulding process, which will cause buckling of the matrices and solvent residual²⁸. Complex and ultra-thin microchannels can be fabricated by template dissolution methods, but dissolution and draining out become challenging due to the capillary effect when the channels are just tens of micrometers^{18, 29}. Other methods, such as employing liquid template³⁰ and laser processing technology³¹, also suffer limitations for 3D geometries and smooth channel generation. In addition, 3D microchannels assembly is also challenging due to templates fixing and removal processes."

To further clarify the above discussion, we added a demonstration for a soft tendril robot containing a 3D helical microchannel structure with a super-high aspect ratio (over 1600=length/diameter = 250 mm/150 μ m), which can achieve winding motion (see Fig.

5b and Supplementary Fig. 6). This super-high aspect ratio helical microchannel hasn't been fabricated before. Moreover, we redesigned the experiments on the artificial vascular model and fabricated a straight artificial vessel (diameter: 150 μm) and a tapered artificial vessel (the minimum diameter: 250 μm , the maximum diameter: 500 μm) in the fragile gel (see Fig. 5e and Supplementary Fig. 6 c-f), which is hard to fabricate by soft lithography.

Specific comments:

RC: (1) With the electrospinning method, smaller diameter products can be produced. Using it as a template, we can create a microchannel by using the melting point difference. Also, I think that a similar microchannel can be fabricated by 3D laser processing. The surface roughness of the inner surface in the microchannel may be superior to our method, but I don't know if there is any application that can take advantage of it.

AR: First of all, this work focuses on the demoulding method. This technique is compatible with most template fabrication methods. Thermal drawing, the template fabrication method used in the paper, is just a typical example. Electrospinning⁶⁰ is, of course, a promising template fabrication method that can be employed in soft demoulding in the future.

Using the melting point difference is a smart method to remove the template, but the viscous polymer melt is hard to drain out of the channels when the channel diameter reaches tens of micrometer since capillary force becomes dominate²⁹. Perhaps with this concern, in most previous literature, the microchannels using templates generated from electrospinning are made by two halves bonded together to facilitate the template removal (e.g., Zeng, Wang et al. 2018), while soft demoulding can create monolithic microchannels. Moreover, the melting method requires a large difference between the melting points of the template and the matrix. Furthermore, with the melting method³⁰, the template can be used only once, while the same template can repeatedly generate similar microchannels in soft demoulding if the template (made from, e.g., TPU) is deformed in the elastic region.

We revised corresponding parts in the 2nd paragraph of "Introduction" section and 2nd paragraph of "Discussion" section as follows:

"The widely accepted soft lithography technique suffers from limited cross-sectional shapes (rectangular) and spatial structures (two-dimensional (2D) patterns only), intensive labour, and expensive fabrication devices, and it is unable to generate monolithic structures,^{6, 19}."

"Other methods, such as employing liquid template³⁰ and laser processing technology³¹, also suffer limitations for 3D geometries and smooth channel generation. In addition, 3D microchannels assembly is also challenging due to templates fixing and removal processes."

"Future research will focus on the improvement of the design (e.g., computer-assisted design⁵⁹), fabrication process (e.g., employing high-precision moving stages for direct ink writing³⁴, and electrospinning technology⁶⁰)."

3D laser processing is also an applicable technology for microchannel fabrication. There are two main types of laser processing: one is subtractive processing and the other is additive processing⁴¹. The subtractive processing utilizes laser thermal ablation to generate microchannels in transparent matrices. Plenty of 3D complex microchannels in transparent and rigid matrices are fabricated by this method (Wang, Yang et al. 2019, Shan, Zhang et al. 2020). Still, the matrix materials are photosensitive glass and fused silica¹¹, which are not suitable for applications on soft devices, such as soft robotics and wearable sensors or artificial tissues. Additive processing was performed by two-photon polymerization to cure the photosensitive resin for 3D micro/nanostructures fabrication. Due to the curing process initiated by a small laser focal point, the surface of the generated microchannel can be rough, and it seems not applicable for large-scale processing. The material is also limited to photosensitive materials.

The smooth surface is important in many applications. For example, soft actuators with high roughness surfaces are susceptible to being broken and obtain a shorter life cycle⁴¹. In optical devices, the microchannels with rough surfaces reduce the transparency and cause more optical intensity loss⁴³. Moreover, in microfluidic valves, the roughness of microchannels influences the fluid interactions⁴² and pressure drop of laminar flow (Lalegani, Saffarian et al. 2018).

We revised the corresponding parts in the 2nd paragraph of "Introduction" section and the 5th paragraph of "Results" section as follows:

"Other methods, such as employing liquid template³⁰ and laser processing technology³¹ also suffer limitations for 3D geometries and smooth channel generation."

"Moreover, the resultant microchannel inner surface is smooth ($S_a = 0.018 \mu\text{m}$) (Supplementary Fig. 2b), which benefits soft robotics, fluidics, and optical applications, such as improving the burst pressure and cycle life of soft actuators⁴¹, enhancing the switching effect of microfluidic valves⁴², and reducing the optical intensity loss for optical waveguides⁴³."

References:

R1. Zeng, J., et al. (2018). "Fabrication of microfluidic channels based on melt-electrospinning direct writing." Microfluidics and Nanofluidics **22**(2): 1-10.

R2. Wang, C., et al. (2019). "Multilayered skyscraper microchips fabricated by hybrid "all-in-one" femtosecond laser processing." Microsystems & Nanoengineering **5**(1): 1-10.

R3. Shan, C., et al. (2020). "Femtosecond laser hybrid fabrication of a 3D microfluidic chip for PCR application." Optics Express **28**(18): 25716-25722.

R4. Lalegani, F., et al. (2018). "Effects of different roughness elements on friction and pressure drop of laminar flow in microchannels." International Journal of Numerical Methods for Heat & Fluid Flow.

RC: (2) Line 24 and line 124: Normally, the aspect ratio of the microchannel is expressed by width-to-depth, but this 6000 is width-to-length. Since it is circular, the aspect ratio is only 1. Adopting your definition, there are many hollow tubes with aspect ratios over 6000 (eg capillary tubes).

AR: As we know, the aspect ratio is a widely accepted index. Its definition varies depending on the fabrication techniques. It is true that in some cases, e.g., photolithography (Miyajima and Mehregany 1995), the aspect ratio is expressed by width-to-depth. In this case, the channel is usually open on one side of the wall, and the depth of the cross section is difficult to be increased. Thus, the width-to-depth ratio can represent the fabrication capabilities. On another aspect, if the channel is monolithic (all walls exist), the aspect ratio is usually defined by the length-to-diameter (Yang, Soper et al. 2007) since the channel's length and diameter are more challenging. In our case, the channel is monolithic, and hence we use the definition of length-to-diameter for the aspect ratio, as shown in Replied Fig. 1.

Therefore, we added the definition of aspect ratio in the 1st paragraph of "Abstract" section as follows:

"The microchannels created by soft demoulding can be as small as 10 μm in diameter and over 6000 in aspect ratio (length-to-diameter)."

Replied Fig. 1 Illustrations of definition of aspect ratio.

About the second question (there are many hollow tubes with aspect ratios over 6000 (e.g., capillary tubes), there are indeed commercially available capillary tubes with a large aspect ratio. They are usually fabricated by extrusion moulding, in which extruding polymer melts or uncured resins through the moulds continually for sheets, films, and pipes fabricating (Kulkarni 2018).

The tube fabricated by this method has two limitations. First, the diameter is limited to several hundred micrometers commercially (see the links in references R4-R6). Second, the cross-section of the channel and the thickness of the wall is usually uniform in the axial direction. In comparison, the microchannel fabricated by soft demoulding is as thin as 10 μm in diameter. Moreover, the fabrication of high-aspect-ratio microchannels is only one trait of soft demoulding. Our soft demoulding technology can also generate 3D complex microchannels in a block matrix.

References:

- R1. Miyajima, H. and M. Mehregany (1995). "High-aspect-ratio photolithography for MEMS applications." Journal of Microelectromechanical Systems **4**(4): 220-229.
- R2. Yang, R., et al. (2007). "A new UV lithography photoresist based on composite of EPON resins 165 and 154 for fabrication of high-aspect-ratio microstructures." Sensors and Actuators A: Physical **135**(2): 625-636.
- R3. Kulkarni, G. S. (2018). Introduction to polymer and their recycling techniques. Recycling of Polyurethane Foams, Elsevier: 1-16.

R4. https://www.spectrumplastics.com/components-technology/extruded-tubing/silicone-extrusion/?gclid=Cj0KCQiAmKiQBhCIARIsAKtSj-kPiE76HTzO45pEV7cKC_Vt6eR_TKvQ6LU8JLR39NHQPqB0pCV44aAaAk_TEALw_wcB

R5. <https://sevitsil.com/medical-grade-silicone-tube.php#target1>

R6. <https://www.raumedic.com/technologies/extrusion/micro-tubing>

RC: (3) Line 85: The authors say "providing great design flexibility," but I don't think it's so flexible because it can only be fabricated by combining soft templates.

AR: The sentence on Line 85 is, "Various thermoplastic polymers are available and adaptable for this manipulation, including low-cost vastly applied polyethylene and polyurethane, providing great design flexibility." Initially, "great design flexibility" means more candidate materials for the template in the fabrication process. We mainly wanted to highlight the broad material range available for the fabrication method. To avoid any confusion caused by "great design flexibility", we deleted this phrase in the 1st paragraph of "Results" section as follows:

"Various thermoplastic polymers are available and adaptable for this manipulation³⁷, including low-cost, vastly applied polyethylene and polyurethane, ~~providing great design flexibility.~~"

RC: a. There are many references where the journal name is not capitalized.

b. Reference 20: Kobayashi, S. is correct.

c. Reference 25: Sensors and Actuators A, 226, 137-142 (2015).

AR: We corrected these typos and errors and proofread the whole draft again.

REVIEWER COMMENTS

Reviewer #1 (Remarks to the Author):

The reviewer thinks that all the comments on the original manuscript have been addressed in this revision. There are two suggestions, both of them on the title of the manuscripts:

(1) The term "Soft demoulding" doesn't seem to capitulate the technological innovation put forward by this work. It is very unfocused, even vague. How about adding the term "self-shrinking" (or any other modifier that the authors think adequate) before or after "soft demoulding"? It would help future readers grasping the concept of the work.

(2) "Ultra-thin" is an obvious exaggeration. The microchannels realized by the proposed method measure 10s to 100s of microns. They are not really considered thin, let along "ultra-thin". The strength of the proposed method is in realizing complex, high-aspect ratio channels. It should be emphasized.

Reviewer #2 (Remarks to the Author):

The authors are under the impression that each comment here has been answered and corrected appropriately in the main. However, I have the impression that the answer to the following RC:(1) is inadequate.

I have the impression that the answer to "I miss a control experiment to show that the vascular model obtained in ~" has not been given. I interpret this as the proposed method's superiority over existing methods not having been shown experimentally. The technical shortcomings of the existing soft lithography are explained, but there seems to be no validation of the performance of the model produced by the proposed technique compared to the existing technique. In addition, it is difficult to read which figure is the model produced by the existing technology. Therefore, it is also difficult to read how much better the model could be compared to the existing technology. For example, if the culture results change between rectangular/cylindrical channels or between channels with/without solvent washing, the impression would be very different.

Reviewer #3 (Remarks to the Author):

I carefully read the revised manuscript and the authors' responses. However, I don't think the authors' processing methods are particularly unique and excellent. I understand the explanation of aspect ratio. However, depending on the processing method, it can be manufactured for as long as possible, and industrially, a larger aspect ratio is realized. I don't understand the appeal of this method.

Title: Self-shrinking soft demoulding for complex high-aspect-ratio microchannels

Authors: Dongliang Fan, Xi Yuan, Wenyu Wu, Renjie Zhu, Xin Yang, Yuxuan Liao, Yunteng Ma, Chufan Xiao, Cheng Chen, Changyue Liu, Hongqiang Wang, Peiwu Qin

Manuscript ID: NCOMMS-21-39450A

Reviewer #1 (Remarks to the Author):

Paper summary: The reviewer thinks that all the comments on the original manuscript have been addressed in this revision.

RC: There are two suggestions, both of them on the title of the manuscripts:

(1) The term "Soft demoulding" doesn't seem to capitulate the technological innovation put forward by this work. It is very unfocused, even vague. How about adding the term "self-shrinking" (or any other modifier that the authors think adequate) before or after "soft demoulding"? It would help future readers grasping the concept of the work.

(2) "Ultra-thin" is an obvious exaggeration. The microchannels realized by the proposed method measure 10s to 100s of microns. They are not really considered thin, let alone "ultra-thin". The strength of the proposed method is in realizing complex, high-aspect-ratio channels. It should be emphasized.

AR: The authors appreciate the reviewer's valuable suggestions.

We revised the title as follows:

"Self-shrinking soft demoulding for complex high-aspect-ratio microchannels"

Reviewer #2:

Paper summary: The authors are under the impression that each comment here has been answered and corrected appropriately in the main. However, I have the impression that the answer to the following RC:(1) is inadequate.

RC: I have the impression that the answer to "I miss a control experiment to show that the vascular model obtained in ~" has not been given. I interpret this as the proposed method's superiority over existing methods not having been shown experimentally. The technical shortcomings of the existing soft lithography are explained, but there seems to be no validation of the performance of the model produced by the proposed technique compared to the existing technique. In addition, it is difficult to read which figure is the model produced by the existing technology. Therefore, it is also difficult to read how much better the model could be compared to the existing technology. For example, if the culture results change between rectangular/cylindrical channels or between channels with/without solvent washing, the impression would be very different.

AR: The authors appreciate the reviewer's comments. We supplemented two control experiments to better exhibit the main advantages (gentle pull-out force and no resident solvent) of our soft demoulding technology over existing technologies. The first one is rigid demoulding for fabricating microchannels in fragile fibrin gels. We embedded a hot melt adhesive filament (diameter: 200 μm), a nylon filament (diameter: 200 μm), and a Nitinol filament (diameter: 300 μm) into a fibrin gel matrix. The large shear force during rigid demoulding for both nylon and Nitinol filaments resulted in rupturing the microchannels in the fibrin gels, as shown in Supplementary Fig. 10a. Soft demoulding using hot melt adhesive filament generates smooth microchannels, as shown in Supplementary Fig. 10a. This control experiment indicates the gentle soft demoulding process is more applicable for fragile gel matrices.

The other control experiment was conducted to test the negative effect on cell growth in artificial vessels resulting from resident solvent, which is employed for both matrix swollen^{16, 22} and template dissolution methods^{18, 25}. We quantitatively tested the effect of the resident solvent by adding acetone (extensively applied for swelling PDMS¹⁶ and dissolving Acrylonitrile-Butadiene-Styrene¹⁸ in the microchannel fabrication processes) into fibrin gels and prepared these gels containing 0%, 0.5%, 1%, and 2% acetone/fibrin gel (v/v). When introducing acetone to the gels, the death rate of the 3T3 cells increased accordingly, as shown in Supplementary Fig. 10b. This control experiment indicates that the resident solvent in matrices causes cell death rate increment; therefore, the solvent-free soft demoulding process is more suitable for biomedical applications.

For the effect of cross-sectional geometry on artificial vessels, plenty of published papers have indicated the superiorities of round microchannels over rectangular ones. The advantages mainly appear in fluid characteristics: (i) the cross-sectional geometry can influence the pressure distribution in a microchannel. Compared to a round cross-section

with a smooth boundary, the shape corner of rectangular geometry tends to generate a higher hydraulic resistance, which may reduce the efficiency of cells or nutrition transporting (Mortensen, Okkels et al. 2005). When large cells pass through a vessel, whose dimension is smaller than the cells, the continuous liquid from four shape corners phase generates high shear force acting on these cells, which may deform and even damage them (Pries, Secomb et al. 1994, Fiddes, Raz et al. 2010, Fenech, Girod et al. 2019); (ii) the cross-sectional geometry can influence the traffic of cells in branched microvessels. For transporting leukocytes into branched channels, the shape transition between rectangular microchannels hinders transporting efficiently, from 69% (round cross section) to 13% (rectangular cross section) (Yang, Forouzan et al. 2011).

The corresponding revision can be found in the 11th paragraph of "Results" section, 17th and 18th paragraphs of "Methods" section, and Supplementary Fig. 10 as follows:

“To exhibit the advantages of soft demoulding, we employed two rigid templates for fabricating microchannels in fragile fibrin gels, but the large shear force caused the microchannels to rupture (see Supplementary Fig. 10a, and "Methods" section 'Fabrication of microchannels in fibrin gels by rigid demoulding'). We also verified the negative effect of acetone, which is employed in the matrix by matrix swollen^{16, 22} and template dissolution methods^{18, 25}, on cell growth in artificial vessels. When introducing acetone to the gels, the death rate of the 3T3 cells increased accordingly, as shown in Supplementary Fig. 10b. Therefore, with the gentle and solvent-free soft demoulding technology for 3D complex microchannel fabrication...”

“To verify the effect of the acetone concentration on the death rate of 3T3 cells, we prepared a series of acetone concentrations (0%, 0.5%, 1%, and 2%, volume ratio) in the fibrin gels. The 3T3 cell concentration in the fibrin gels was 1.5×10^6 cell/mL. The artificial vessels were stained with a fluorescent live/dead assay after 2 days of culture, as shown in Supplementary Fig. 10b.

Fabrication of microchannels in fibrin gels by rigid demoulding. We first prepared two rigid templates, a nylon filament (diameter: 200 μ m) and a Nitinol filament (diameter: 300 μ m), for rigid demoulding. The microchannels in the fibrin gels were fabricated by following the same protocol shown in Supplementary Fig. 1. The 3T3 cell concentration in the fibrin gels was 5×10^5 cell/mL, and the fabricated microchannels are shown in Supplementary Fig. 10a.”

The reviewer also mentioned that “it is difficult to read which figure is the model produced by the existing technology.” In order to avoid confusion, we add one sentence to the legend of Figure 5 to describe all artificial vessels presented in Figure 5e are fabricated by soft demoulding, as follows:

“e, The artificial blood vessels in fibrin gels with HUVECs seeded, fabricated by soft demoulding.”

Supplementary Fig. 10 Effect of rigid demoulding and solvent resident on biomedical applications. **a**, The microchannels fabricated by hot melt adhesive filament (intact), Nitinol filament (ruptured), and nylon filament (ruptured) from left to right. Scale bars, 200 μm . **b**, Effect of different acetone concentrations on the death rate of cells. The left ones are brightfield images, and the middle (dead cells) and right ones (dead and live cells) are confocal images of the fibrin gel after two days of culture stained with live (green)/dead (red) assay. Scale bars, 200 μm .

References

R1: Mortensen, N. A., et al. (2005). "Reexamination of Hagen-Poiseuille flow: Shape dependence of the hydraulic resistance in microchannels." Physical Review E **71**(5): 057301.

R2: Pries, A. R., et al. (1994). "Resistance to blood flow in microvessels in vivo." Circulation Research **75**(5): 904-915.

R3: Fenech, M., et al. (2019). "Microfluidic blood vasculature replicas using backside lithography." Lab on a Chip **19**(12): 2096-2106.

R4: Fiddes, L. K., et al. (2010). "A circular cross-section PDMS microfluidics system for replication of cardiovascular flow conditions." Biomaterials **31**(13): 3459-3464.

R5: Yang, X., et al. (2011). "Traffic of leukocytes in microfluidic channels with rectangular and rounded cross-sections." Lab on a Chip **11**(19): 3231-3240.

Reviewer #3:

RC: I carefully read the revised manuscript and the authors' responses. However, I don't think the authors' processing methods are particularly unique and excellent. I understand the explanation of aspect ratio. However, depending on the processing method, it can be manufactured for as long as possible, and industrially, a larger aspect ratio is realized. I don't understand the appeal of this method.

AR: We feel regret that we cannot reach an agreement on the novelty of our soft demoulding technology, but the authors still would like to thank the reviewer for your valuable time and frank comments on the manuscript. We want to mention that although generating high-aspect-ratio microchannels is already realized commercially, however, the channel geometry is simply straight, and the dimension is several hundred microns, as we mentioned in the last response letter. We want to emphasize that our soft demoulding is not only capable of fabricating a straight microchannel with a high aspect ratio (over 6000) but a 3D helical microchannel in a soft matrix (e.g., the soft tendril robot with an aspect ratio over 1600). Combined with moulding and soft template placement, different intricate microchannel patterns are produced to demonstrate the fabrication ability for 3D complex microchannels. Moreover, according to the theoretical model for soft demoulding, our fabrication method is not only applicable for generating microchannels in soft matter, such as rubbers and silicones, but also in rigid materials, such as resins, ceramics, and glasses, which exhibit much broader applicable regions, compared to commercialized microchannel fabrication technologies.

Reviewers' Comments:

Reviewer #2:

Remarks to the Author:

The authors fully addressed the reviewer concerns.

Title: Self-shrinking soft demoulding for complex high-aspect-ratio microchannels

Authors: Dongliang Fan, Xi Yuan, Wenyu Wu, Renjie Zhu, Xin Yang, Yuxuan Liao, Yunteng Ma, Chufan Xiao, Cheng Chen, Changyue Liu, Hongqiang Wang, Peiwu Qin

Manuscript ID: NCOMMS-21-39450B

Reviewer #2 (Remarks to the Author):

RC: The authors fully addressed the reviewer concerns.

AR: The authors appreciate the reviewer's valuable suggestions during the review.